# Learning Fine-grained View-Invariant Representations from Unpaired Ego-Exo Videos via Temporal Alignment

Zihui Xue[1,2]    Kristen Grauman[1,2]
[1]The University of Texas at Austin    [2]FAIR, Meta

## Abstract

The egocentric and exocentric viewpoints of a human activity look dramatically different, yet invariant representations to link them are essential for many potential applications in robotics and augmented reality. Prior work is limited to learning view-invariant features from *paired* synchronized viewpoints. We relax that strong data assumption and propose to learn fine-grained action features that are invariant to the viewpoints by aligning egocentric and exocentric videos in time, even when not captured simultaneously or in the same environment. To this end, we propose AE2, a self-supervised embedding approach with two key designs: (1) an object-centric encoder that explicitly focuses on regions corresponding to hands and active objects; and (2) a contrastive-based alignment objective that leverages temporally reversed frames as negative samples. For evaluation, we establish a benchmark for fine-grained video understanding in the ego-exo context, comprising four datasets—including an ego tennis forehand dataset we collected, along with dense per-frame labels we annotated for each dataset. On the four datasets, our AE2 method strongly outperforms prior work in a variety of fine-grained downstream tasks, both in regular and cross-view settings.[1]

## 1   Introduction

*Fine-grained video understanding* aims to extract the different stages of an activity and reason about their temporal evolution. For example, whereas a coarse-grained action recognition task [31, 16, 68, 20, 7] might ask, "is this sewing or snowboarding or...?", a fine-grained action recognition task requires detecting each of the component steps, *e.g.*, "the person threads the needle here, they poke the needle through the fabric here..." Such fine-grained understanding of video is important in numerous applications that require step-by-step understanding, ranging from robot imitation learning [42, 63] to skill learning from instructional "how-to" videos [86, 73, 47, 46].

The problem is challenging on two fronts. First, the degree of detail and precision eludes existing learned video representations, which are typically trained to produce global, clip-level descriptors of the activity [69, 74, 18, 5]. Second, in many scenarios of interest, the camera viewpoint and background will vary substantially, making the same activity look dramatically different. For example, imagine a user wearing AR/VR glasses who wants to compare the egocentric recording of themselves doing a tennis forehand with a third-person (exocentric) expert demonstration video on YouTube, or a robot tasked with pouring drinks from its egocentric perspective but provided with multiple third-person human pouring videos for learning. Existing view-invariant feature learning [76, 27, 78, 45, 57, 58]—including methods targeting ego-exo views [63, 66]—assume that training data contains synchronized video pairs capturing the same action *simultaneously in the same environment*. This is a strong assumption. It implies significant costs and physical constraints that can render these methods inapplicable in complex real-world scenarios.

---

[1]Project webpage: `https://vision.cs.utexas.edu/projects/AlignEgoExo/`.

37th Conference on Neural Information Processing Systems (NeurIPS 2023).

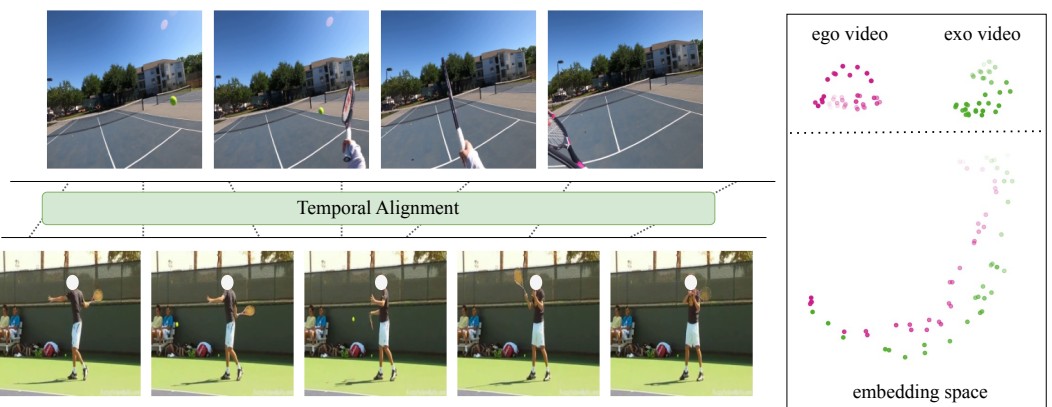

Figure 1: Left: We propose AE2, a self-supervised approach for learning frame-wise action features that are invariant to the egocentric and exocentric viewpoints. The key is to find correspondences across the viewpoints via temporal alignment. Right: Before training, the frame embeddings are clustered by viewpoint (pink and green); after training, our learned AE2 embeddings effectively capture the progress of an action and exhibit viewpoint invariance. Shading indicates time.

Our goal is to learn fine-grained frame-wise video features that are invariant to both the ego and exo views,[2] from *unpaired* data. In the unpaired setting, we know which human activity occurs in any given training sequence (*e.g.*, pouring, breaking eggs), but they need not be collected simultaneously or in the same environment. The main idea of our self-supervised approach is to infer the temporal alignment between ego and exo training video sequences, then use that alignment to learn a view-invariant frame-wise embedding. Leveraging unpaired data means our method is more flexible—both in terms of curating training data, and in terms of using available training data more thoroughly, since with our approach *any* ego-exo pairing for an action becomes a training pair. Unlike existing temporal alignment for third-person videos [15, 54, 22, 23, 41], where canonical views are inherently similar, ego-exo views are so distinct that finding temporal visual correspondences is non-trivial (see Fig. 1).

To address this challenge, we propose two main ideas for aligning ego and exo videos in time. First, motivated by the fact that human activity in the first-person view usually revolves around hand-object interactions, we design an object-centric encoder to bridge the gap between ego-exo viewpoints. The encoder integrates regional features corresponding to hands and active objects along with global image features, resulting in more object-centric representations. Second, we propose a contrastive-based alignment objective, where we temporally reverse frames to construct negative samples. The rationale is that aligning an ego-exo video pair should be easier (*i.e.*, incur a lower alignment cost) compared to aligning the same video pair when one is played in reverse. We bring both ideas together in a single embedding learning approach we call AE2, for "align ego-exo".

To evaluate the learned representations, we establish an ego-exo benchmark for fine-grained video understanding. Specifically, we assemble four datasets of atomic human actions, comprising ego and exo videos drawn from five public datasets and an ego tennis forehand dataset that we collected. In addition, we annotate these datasets with dense per-frame labels (for evaluation only). We hope that the enriched data and labels that we release publicly can support progress in fine-grained temporal understanding from the ego-exo perspective. This is a valuable dataset contribution alongside our technical approach contribution. Furthermore, we present a range of practical downstream tasks that demand frame-wise ego-exo view-invariant features: action phase classification and retrieval across views, progress tracking, and synchronous playback of ego and exo videos. Experimental results demonstrate considerable performance gains of AE2 across all tasks and datasets.

## 2    Related Work

**Fine-grained Action Understanding**    To recognize fine-grained actions, supervised learning approaches [82, 30, 59, 65] employ fine-grained action datasets annotated with sub-action boundaries. In contrast, our self-supervised approach requires much lighter annotations to pretrain for

---

[2]We use "ego" and "exo" as shorthand for egocentric (first-person) and exocentric (third-person).

action phase classification, namely, video-level action labels. Keystep recognition in instructional videos [86, 73, 47, 46, 2] is a related form of fine-grained action understanding, where the goal is to name the individual steps of some longer activity (*e.g.*, making brownies requires breaking eggs, adding flour, etc.). In that domain, multi-modal representation learning is a powerful tool that benefits from the spoken narrations that accompany how-to videos [47, 72, 38, 84]. In any of the above, handling extreme viewpoint variation is not an explicit target.

**Self-supervised Video Representation Learning**  Various objectives for self-supervised video features have been explored [61]. This includes using accompanying modalities like audio [51, 29, 49] and text or speech [47, 72, 38, 37, 84] as training signals, as well as temporal coherence priors [28, 26, 63, 44, 10] or self-supervised tasks that artificially alter the temporal order of videos [48, 19, 35]. Alignment-based objectives have also been explored [15, 54, 22, 23, 41, 33], with ideas based on cycle consistency [15, 54], soft dynamic time warping (DTW) [22, 23], and optimal transport (OT) [41]. Our model also has an alignment stage; however, unlike our work, all these prior methods are explored for third-person videos only (where views are generally more similar across videos than ego-exo) without attending to the viewpoint discrepancy problem. Frame-wise representations invariant to both ego and exo viewpoints remain unresolved.

**View-invariant Action Representation Learning**  Many prior works have explored view-invariance in action recognition [76, 9, 53]. One popular approach is to employ multi-view videos during training to learn view-invariant features [76, 77, 27, 78, 45, 71]. Several works [17, 39, 83, 55] discover a latent feature space independent of viewpoint, while others use skeletons [81], 2D/3D poses [56, 71] and latent 3D representations [53] extracted from video to account for view differences.

As advances in AR/VR and robotics unfold, ego-exo viewpoint invariance takes on special interest. Several models relate coarse-grained action information across the two drastically different domains [70, 1, 80, 36], for action recognition using *both* ego and exo cameras [70] or cross-viewpoint video recognition [80, 1, 36]. Closer to our work are Time Contrastive Networks (TCN) [63] and CharadesEgo [66, 67], both of which use embeddings aimed at pushing (ir)relevant ego-exo views closer (farther). However, unlike our approach, both employ videos that *concurrently* record the same human actions from ego and exo viewpoints. In fact, except for [36], all the above approaches rely on simultaneously recorded, paired multi-view videos. However, even in the case of [36], the goal is a coarse, clip-wise representation. In contrast, our method leverages temporal alignment as the pretext task, enabling fine-grained representation learning from *unpaired* ego and exo videos.

**Object-centric Representations**  Several works advocate object-centric representations to improve visual understanding [40, 85, 50, 36], such as an object-centric transformer for joint hand motion and interaction hotspot prediction [40], or using a pretrained detection model for robot manipulation tasks [85, 50]. Hand-object interactions are known to dominate ego-video for human-worn cameras [11, 21], and recent work explores new ways to extract hands and active objects [64, 43, 12]. Our proposed object-centric transformer shares the rationale of focusing on active objects, though in our case as a key to link the most "matchable" things in the exo view and bridge the ego-exo gap.

## 3 Approach

We first formulate the problem (Sec. 3.1) and introduce the overall self-supervised learning objective (Sec. 3.2). Next we present our ideas for object-centric transformer (Sec. 3.3) and an alignment-based contrastive regularization (Sec. 3.4), followed by a review of our training procedure (Sec. 3.5).

### 3.1  Learning Frame-wise Ego-Exo View-invariant Features

Our goal is to learn an embedding space shared by both ego and exo videos that characterizes the progress of an action, as depicted in Fig. 1 (right). Specifically, we aim to train a view-invariant encoder network capable of extracting fine-grained frame-wise features from a given (ego or exo) video. Training is conducted in a self-supervised manner.

Formally, let $\phi(\cdot; \theta)$ represent a neural network encoder parameterized by $\theta$. Given an ego video sequence $\mathbf{S}$ with $M$ frames, denoted by $\mathbf{S} = [\mathbf{s}_1, \ldots, \mathbf{s}_M]$, where $\mathbf{s}_i$ denotes the $i$-th input frame, we apply the encoder to extract frame-wise features from $\mathbf{S}$, *i.e.*, $\mathbf{x}_i = \phi(\mathbf{s}_i; \theta), \forall i \in \{1, \ldots, M\}$. The resulting ego embedding is denoted by $\mathbf{X} = [\mathbf{x}_1, \ldots, \mathbf{x}_M]$. Similarly, given an exo video sequence $\mathbf{V} = [\mathbf{v}_1, \ldots, \mathbf{v}_N]$ with $N$ frames, we obtain the frame-wise exo embedding $\mathbf{Y} = [\mathbf{y}_1, \ldots, \mathbf{y}_N]$, where $\mathbf{y}_i = \phi(\mathbf{v}_i; \theta), \forall i \in \{1, \ldots, N\}$ using the same encoder $\phi$.

The self-supervised learning objective is to map $\mathbf{S}$ and $\mathbf{V}$ to a shared feature space where frames representing similar action stages are grouped together, regardless of the viewpoint, while frames depicting different action stages (*e.g.*, the initial tilting of the container versus the subsequent decrease in the liquid stream as the pouring action nears completion) should be separated. We present a series of downstream tasks to evaluate the quality of learned feature embeddings $\mathbf{X}$ and $\mathbf{Y}$ (Sec. 4).

An intuitive approach is to utilize paired ego-exo data collected simultaneously, adopting a loss function that encourages similarities between frame representations at the same timestamp while maximizing the distance between frame representations that are temporally distant [63, 66]. However, this approach relies on time-synchronized views, which is often challenging to realize in practice.

Instead, we propose to achieve view-invariance from *unpaired* ego-exo data; the key is to adopt temporal alignment as the pretext task. The encoder $\phi$ learns to identify visual correspondences across views by temporally aligning ego and exo videos that depict the same action. Throughout the learning process, no action phase labels or time synchronization is required. To enforce temporal alignment, we assume that multiple videos portraying the same action (*e.g.*, pouring, tennis forehand), captured from different ego and exo viewpoints (and potentially entirely different environments), are available for training.

## 3.2 Ego-Exo Temporal Alignment Objective

To begin, we pose our learning objective as a classical dynamic time warping (DTW) problem [4]. DTW has been widely adopted to compute the optimal alignment between two temporal sequences [8, 6, 23, 22, 79] according to a predefined cost function. In our problem, we define the cost of aligning frame $i$ in ego sequence $\mathbf{S}$ and frame $j$ in exo sequence $\mathbf{V}$ as the distance between features extracted by our encoder $\phi$, $\mathbf{x}_i$ and $\mathbf{y}_j$, so that aligning more similar features results in a lower cost.

Formally, given the two sequences of embeddings $\mathbf{X}$ of length $M$ and $\mathbf{Y}$ of length $N$, we calculate a cost matrix $\mathbf{C} \in \mathbb{R}^{M \times N}$, with each element $c_{i,j}$ computed by a distance function $\mathcal{D}$. We adopt the distance function from [22], and define $c_{i,j}$ to be the negative log probability of matching $\mathbf{x}_i$ to a selected $\mathbf{y}_j$ in sequence $\mathbf{Y}$:

$$c_{i,j} = \mathcal{D}(\mathbf{x}_i, \mathbf{y}_j) = -\log \frac{\exp(\tilde{\mathbf{x}}_i^T \tilde{\mathbf{y}}_j / \beta)}{\sum_{k=1}^N \exp(\tilde{\mathbf{x}}_i^T \tilde{\mathbf{y}}_k / \beta)}, \quad \tilde{\mathbf{x}}_i = \mathbf{x}_i / \|\mathbf{x}_i\|_2 \quad \tilde{\mathbf{y}}_i = \mathbf{y}_i / \|\mathbf{y}_i\|_2 \quad (1)$$

where $\beta$ is a hyper-parameter controlling the softmax temperature, which we fix at 0.1.

DTW calculates the alignment cost between $\mathbf{X}$ and $\mathbf{Y}$ by finding the minimum cost path in $\mathbf{C}$. This optimal path can be determined by evaluating the following recurrence [60]:

$$\mathbf{R}(i,j) = c_{i,j} + \min[\mathbf{R}(i-1, j-1), \mathbf{R}(i-1, j), \mathbf{R}(i, j-1)] \quad (2)$$

where $\mathbf{R}(i,j)$ denotes the cumulative distance function, and $\mathbf{R}(0,0) = 0$. Given that the min operator is non-differentiable, we approximate it with a smooth differentiable operator, as defined in [34, 22]. This yields a differentiable loss function that can be directly optimized with back-propagation:

$$\mathcal{L}_{\text{align}}(\mathbf{X}, \mathbf{Y}) = \mathbf{R}(M, N). \quad (3)$$

While DTW has been applied in frame-wise action feature learning [23, 22] with some degree of view invariance across exo views, it is inadequate to address the ego-exo domain gap. As we will show in results, directly adopting the DTW loss for learning ego-exo view-invariant features is sub-optimal since the ego and exo views, even when describing the same action, exhibit significant differences (compared to differences within ego views or within exo views). In Sec. 3.3-3.4, we propose two designs of AE2 that specifically account for the dramatic ego-exo view differences.

## 3.3 Object-centric Encoder Network

The visual appearance in ego and exo views may look dramatically different due to different backgrounds, fields of view, and (of course) widely disparate viewing angles. Yet a common element shared between these two perspectives is the human action itself. In this context, *hand-object interaction regions* are particularly informative about the progress of an action. Hand-object interactions are prominent in the first-person viewpoint; the wearable camera gives a close-up view of the person's near-field manipulations. In exo videos, however, these active regions may constitute only a small

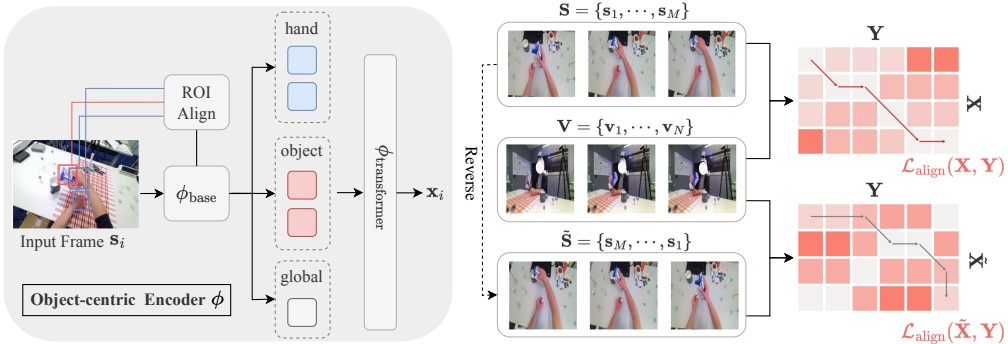

Figure 2: Overview of AE2. We extract frame-wise representations from the two video sequences (*i.e.*, $\mathbf{S}$ and $\mathbf{V}$) using the encoder $\phi$ and temporally align their embeddings (*i.e.*, $\mathbf{X}$ and $\mathbf{Y}$) using DTW. Left: Our proposed encoder network design emphasizes attention to hand and active object regions, leading to more object-centric representations. Right: As a form of regularization, we reverse the video sequence $\mathbf{S}$ and enforce that the cost of aligning $(\mathbf{S}, \mathbf{V})$ is less than that of aligning $(\tilde{\mathbf{S}}, \mathbf{V})$.

portion of the frame and are not necessarily centered, depending on the third-person camera position and angle. Inspired by these observations, we propose an object-centric encoder design that explicitly focuses on regions corresponding to hands and active objects to better bridge the ego-exo gap.

Our proposed AE2 encoder network $\phi$ consists of two main components: (1) a ResNet-50 encoder [25], denoted by $\phi_{\text{base}}$, to extract features from the input video frame; (2) a transformer encoder [75], denoted by $\phi_{\text{transformer}}$, to fuse global features along with regional features through self-attention.

As a preprocessing step, we employ an off-the-shelf detector to localize hand and active object regions in ego and exo frames [64]. For frame $i$, we select the top two hand and object proposals with the highest confidence scores, resulting in a total of four bounding boxes $\{\text{BB}_{i,k}\}_{k=1}^4$.

First, given one input frame $\mathbf{s}_i$ at timestamp $i$, we pass it to $\phi_{\text{base}}$ to obtain global features: $\mathbf{g}_i = \phi_{\text{base}}(\mathbf{s}_i)$. Next, we apply ROI Align [24] to intermediate feature maps produced by $\phi_{\text{base}}$ for the input frame $\mathbf{s}_i$. This yields regional features $\mathbf{r}_{i,k}$ for each bounding box $\text{BB}_{i,k}$: $\mathbf{r}_{i,k} = \text{ROI Align}(\phi_{\text{base}}^{\text{interm.}}(\mathbf{s}_i), \text{BB}_{i,k}), \forall k$.

Both the global and regional features are flattened as vectors and passed through a projection layer to unify their dimensions. We denote the projection layer to map global and local tokens as $\phi_{\text{project}_g}$ and $\phi_{\text{project}_l}$, respectively. The input tokens are then enriched with two types of embeddings: (1) a learnable spatial embedding, denoted by $\mathbf{e}_{\text{spatial}}$, which encodes the bounding box coordinates and confidence scores obtained from the hand-object detector and (2) a learnable identity embedding, denoted by $\mathbf{e}_{\text{identity}}$, which represents the category of each feature token, corresponding to the global image, left hand, right hand, and object. This yields the tokens: $\mathbf{z}_{i,\text{global}} = \phi_{\text{project}_g}(\mathbf{g}_i) + \mathbf{e}_{\text{identity}}$ and $\mathbf{z}_{i,\text{local}k} = \phi_{\text{project}_l}(\mathbf{r}_{i,k}) + \mathbf{e}_{\text{spatial}} + \mathbf{e}_{\text{identity}}, \forall k$.

The transformer encoder processes one global token along with four regional tokens as input, leveraging self-attention to conduct hand-object interaction reasoning. This step enables the model to effectively capture action-related information within the scene, yielding the transformer outputs: $\mathbf{f}_{i,k} = \phi_{\text{transformer}}(\mathbf{z}_{i,\text{global}}, \{\mathbf{z}_{i,\text{local}k}\}_k)$. Finally, we average the transformer output tokens and adopt one embedding layer $\phi_{\text{embedding}}$ to project the features to our desired dimension, yielding the final output embedding for the input frame $\mathbf{s}_i$: $\mathbf{x}_i = \phi_{\text{embedding}}\left(\frac{1}{5}\sum_k \mathbf{f}_{i,k}\right)$.

Fig. 2 (left) showcases our proposed object-centric encoder. This design inherently bridges the ego-exo gap by concentrating on the most informative regions, regardless of the visual differences between the two perspectives. As a result, the object-centric encoder can better align features extracted from both ego and exo videos, enhancing the learned features in downstream tasks.

### 3.4 Contrastive Regularization With Reversed Videos

As shown in previous studies [22, 23], directly optimizing the DTW loss can result in the collapse of embeddings and the model settling into trivial solutions. Moreover, aligning ego and exo videos

presents new challenges due to their great differences in visual appearance. Thus, a careful design of regularization, supplementing the alignment objective in Eqn. (3), is imperative.

We resort to discriminative modeling as the regularization strategy. The crux lies in how to construct positive and negative samples. A common methodology in previous fine-grained action approaches [63, 66, 23, 10], is to define negative samples on a frame-by-frame basis: temporally close frames are considered as positive pairs (expected to be nearby in the embedding space), while temporally far away frames are negative pairs (expected to be distant). This, however, implicitly assumes clean data and strict monotonicity in the action sequence: if a video sequence involves a certain degree of periodicity, with many visually similar frames spread out over time, these approaches could incorrectly categorize them as negative samples.

To address the presence of noisy frames in real-world videos, we introduce a global perspective that creates negatives at the video level. From a given positive pair of sequences $(\mathbf{S}, \mathbf{V})$, we derive a negative pair $(\tilde{\mathbf{S}}, \mathbf{V})$, where $\tilde{\mathbf{S}} = [\mathbf{s}_M, \ldots, \mathbf{s}_1]$ denotes the *reversed* video sequence $\mathbf{S}$. The underlying intuition is straightforward: aligning an ego-exo video pair in their natural temporal order should yield a lower cost than aligning the same pair when one video is played in reverse. Formally, we employ a hinge loss to impose this regularization objective:

$$\mathcal{L}_{\text{reg}}(\mathbf{X}, \mathbf{Y}) = \max(\mathcal{L}_{\text{align}}(\mathbf{X}, \mathbf{Y}) - \mathcal{L}_{\text{align}}(\tilde{\mathbf{X}}, \mathbf{Y}), 0) \quad (4)$$

where $\tilde{\mathbf{X}}$ denotes the embeddings of $\tilde{\mathbf{S}}$, which is essentially $\mathbf{X}$ in reverse. Fig. 2 (right) illustrates the alignment cost matrices of $(\mathbf{X}, \mathbf{Y})$ and $(\tilde{\mathbf{X}}, \mathbf{Y})$. A lighter color indicates a smaller cost. Intuitively, $\mathcal{L}_{\text{align}}$ corresponds to the minimal cumulative cost traversing from the top-left to the bottom-right of the matrix. As seen in the figure, the alignment cost increases when $\mathbf{X}$ is reversed, *i.e.*, $\mathcal{L}_{\text{align}}(\tilde{\mathbf{X}}, \mathbf{Y})$ should be larger than $\mathcal{L}_{\text{align}}(\mathbf{X}, \mathbf{Y})$.

Compared with frame-level negatives, our formulation is inherently more robust to repetitive and background frames, allowing for a degree of temporal variation within a video. This principle guides two key designs: (1) In creating negative samples, we opt for reversing the frames in $\mathbf{S}$ rather than randomly shuffling them. The latter approach may inadvertently generate a plausible (hence, not negative) temporal sequence, particularly when dealing with short videos abundant in background frames. (2) To obtain informative frames as the positive sample, we leverage the hand-object detection results from Sec. 3.3 to perform weighted sampling of the video sequences $\mathbf{S}$ and $\mathbf{V}$. Specifically, we sample frames proportionally to their average confidence scores of hand and object detections. The subsampled $\mathbf{S}$ and $\mathbf{V}$ thus compose a positive pair. This way, frames featuring clear hand and object instances are more likely to be included.

### 3.5 Training and Implementation Details

Our final loss is a combination of the DTW alignment loss (Eqn. (3)) and the contrastive regularization loss (Eqn. (4)):

$$\mathcal{L}(\mathbf{X}, \mathbf{Y}) = \mathcal{L}_{\text{align}}(\mathbf{X}, \mathbf{Y}) + \lambda \mathcal{L}_{\text{reg}}(\mathbf{X}, \mathbf{Y}), \quad (5)$$

where $\lambda$ is a hyper-parameter controlling the ratio of these two loss terms.

During training, we randomly extract 32 frames from each video to construct a video sequence. The object-centric encoder network $\phi$, presented in Sec. 3.3, is optimized to minimize the loss above for all pairs of video sequences in the training set, including ego-ego, exo-exo, and ego-exo pairs[3]—all of which are known to contain the same coarse action. During evaluation, we freeze the encoder $\phi$ and use it to extract 128-dimensional embeddings for each frame. These representations are then assessed across a variety of downstream tasks (Sec. 4). See Supp. for full implementation details.

## 4 Datasets and Evaluation

**Datasets** Existing datasets for fine-grained video understanding (*e.g.* [82, 65, 3]) are solely composed of third-person videos. Hence we propose a new ego-exo benchmark by assembling ego and exo videos from five public datasets—CMU-MMAC [13], H2O [32], EPIC-Kitchens [11], HMDB51 [31]

---

[3]In Sec. 3.1, we initially denote $\mathbf{X}$ and $\mathbf{Y}$ as ego and exo embeddings. However, in this context, we extend the meaning of $\mathbf{X}$ and $\mathbf{Y}$ to represent any pair of embeddings from the training set.

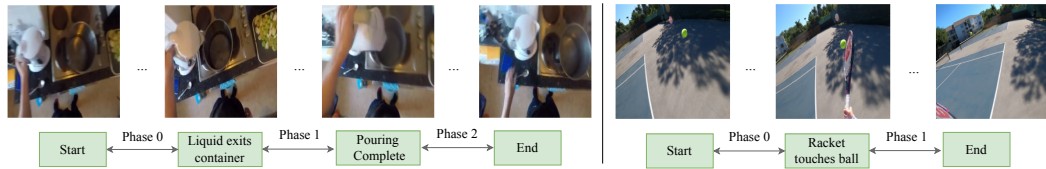

Figure 3: Example labels for Pour Liquid (left) and Tennis Forehand (right). Key events are displayed in boxes below sequences, with the phase label assigned to each frame between two key events.

and Penn Action [82]—plus a newly collected ego tennis forehand dataset. Our benchmark consists of four action-specific ego-exo datasets: (A) **Break Eggs**: ego-exo videos of 44 subjects breaking eggs, with 118/30 train/val+test samples from CMU-MMAC [13]; (B) **Pour Milk** [32]: ego-exo videos of 10 subjects pouring milk, with 77/35 train/val+test from H2O [32]; (C) **Pour Liquid**: we extract ego "pour water" clips from EPIC-Kitchens [11] and exo "pour" clips from HMDB51 [31], yielding 137/56 train/val+test. (D) **Tennis Forehand**: we extract exo tennis forehand clips from Penn Action [82] and collect ego tennis forehands by 12 subjects, totaling 173/149 train/val+test clips. We obtain **ground truth key event and action phase labels** for all of them (used only in downstream evaluation), following the labeling methodology established in [15]. Fig. 3 illustrates examples of these labels. Altogether, these four datasets provide unpaired ego-exo videos covering both tabletop (*i.e.*, breaking eggs, pouring) and physical actions (*i.e.*, tennis forehand).[4] While (A) and (B) are from more controlled environments, where exo viewpoints are fixed and both views are collected in the same environment, (C) and (D) represent in-the-wild scenarios, where ego and exo are neither synchronized nor collected in the same environment. Our model receives them all as unpaired.

**Evaluation** We aim to demonstrate the practical utility of the learned ego-exo view-invariant features by addressing two questions: (1) how well do the features capture the progress of the action? (2) how view-invariant are the learned representations? To this end, we test four downstream tasks: (1) **Action Phase Classification**: train an SVM classifier on top of the embeddings to predict the action phase labels for each frame and report F1 score. We also explore few-shot and cross-view zero-shot settings (*i.e.*, "exo2ego" when training the SVM classifier on exo only and testing on ego, and vice versa). (2) **Frame Retrieval**: parallels the classification task, but done via retrieval (no classifier), and evaluated with mean average precision (mAP)$@K$, for $K$=5,10,15. We further extend this task to the cross-view setting to evaluate view-invariance. (3) **Phase Progression**: train a linear regressor on the frozen embeddings to predict the phase progression values, defined as the difference in time-stamps between any given frame and each key event, normalized by the number of frames in that video [15], evaluated by average R-square. (4) **Kendall's Tau**: to measure how well-aligned two sequences are in time, following [15, 22, 23, 10, 41].

Note that: (1) Departing from the common practice of splitting videos into train and validation only [15], our benchmark includes a dedicated test set to reduce the risk of model overfitting. (2) We opt for the F1 score instead of accuracy for action phase classification to better account for the label imbalance in these datasets and provide a more meaningful performance evaluation. (3) Phase progression assumes a high level of consistency in actions, with noisy frames diminishing the performance greatly. Due to the challenging nature of Pour Liquid data, we observe a negative progression value for all approaches. Thus, we augment the resulting embeddings with a temporal dimension, as 0.001 times the time segment as the input so that the regression model can distinguish repetitive (or very similar) frames that differ in time. We report modified progression value for all approaches on this dataset. (4) Kendall's Tau assumes that there are no repetitive frames in a video. Since we adopt in-the-wild videos where strict monoticity is not guaranteed, this metric may not faithfully reflect the quality of representations. Nonetheless, we report them for completeness and for consistency with prior work. (5) With our focus being frame-wise representation learning, the total number of training samples equals the number of frames rather than the number of videos, reaching a scale of a few thousands (8k-30k per action class). See Supp. for more dataset and evaluation details.

## 5 Experiments

**Baselines** We compare AE2 with 8 total baselines of three types: (1) Naive baselines based on **random features** or **ImageNet features**, produced by a ResNet-50 model randomly initialized

---

[4]Although Break Eggs (and a portion of Pour Milk) offers synchronized ego-exo pairs, AE2 ignores the pairing and is trained in a self-supervised manner. AE2 never uses this synchronization as a supervision signal.

Table 1: Benchmark evaluation on four datasets: (A) Break Eggs; (B) Pour Milk; (C) Pour Liquid; (D) Tennis Forehand. Top results are highlighted in **bold**, while second-best results are underlined. AE2 performs best across all tasks and datasets, in both regular and cross-view scenarios.

| Dataset | Method | Classification (F1 score) | | | Frame Retrieval (mAP@10) | | | Phase Progression | Kendall's Tau |
|---|---|---|---|---|---|---|---|---|---|
| | | regular | ego2exo | exo2ego | regular | ego2exo | exo2ego | | |
| (A) | Random Features | 19.18 | 18.93 | 19.45 | 47.13 | 41.74 | 38.19 | -0.0572 | 0.0018 |
| | ImageNet Features | 50.24 | 21.48 | 32.25 | 50.49 | 33.09 | 37.80 | -0.1446 | 0.0188 |
| | ActorObserverNet [66] | 36.14 | 36.40 | 31.00 | 50.47 | 42.70 | 41.29 | -0.0517 | 0.0024 |
| | single-view TCN [63] | 56.90 | 18.60 | 35.61 | 53.42 | 32.63 | 34.91 | 0.0051 | 0.1206 |
| | multi-view TCN [63] | 59.91 | 48.65 | 56.91 | 58.83 | 47.04 | 52.68 | 0.2669 | 0.2886 |
| | CARL [10] | 43.43 | 28.35 | 29.22 | 46.04 | 37.38 | 39.94 | -0.0837 | -0.0091 |
| | TCC [15] | 59.84 | 54.17 | 52.28 | 58.75 | 61.11 | 62.03 | 0.2880 | 0.5191 |
| | GTA [22] | 56.86 | 52.33 | 58.35 | 61.55 | 56.25 | 53.93 | 0.3462 | 0.4626 |
| | AE2 (ours) | **66.23** | **57.41** | **71.72** | **65.85** | **64.59** | **62.15** | **0.5109** | **0.6316** |
| (B) | Random Features | 36.84 | 33.96 | 41.97 | 52.48 | 50.56 | 51.98 | -0.0477 | 0.0050 |
| | ImageNet Features | 41.59 | 39.93 | 45.52 | 54.09 | 27.31 | 43.21 | -2.6681 | 0.0115 |
| | single-view TCN [63] | 47.39 | 43.44 | 42.28 | 57.00 | 46.48 | 47.20 | -0.3238 | -0.0197 |
| | CARL [10] | 48.79 | 52.41 | 43.01 | 55.01 | 52.99 | 51.51 | -0.1639 | 0.0443 |
| | TCC [15] | 77.91 | 72.29 | 81.07 | 80.97 | 75.30 | 80.27 | 0.6665 | 0.7614 |
| | GTA [22] | 81.11 | 74.94 | 81.51 | 80.12 | 72.78 | 75.40 | 0.7086 | 0.8022 |
| | AE2 (ours) | **85.17** | **84.73** | **82.77** | **84.90** | **78.48** | **83.41** | **0.7634** | **0.9062** |
| (C) | Random Features | 45.26 | 47.45 | 44.33 | 49.83 | 55.44 | 55.75 | -0.1303 | -0.0072 |
| | ImageNet Features | 53.13 | 22.44 | 44.61 | 51.49 | 52.17 | 30.44 | -1.6329 | -0.0053 |
| | single-view TCN [63] | 54.02 | 32.77 | 51.24 | 48.83 | 55.28 | 31.15 | -0.5283 | 0.0103 |
| | CARL [10] | 56.98 | 47.46 | 52.68 | 55.29 | 59.37 | 36.80 | -0.1176 | 0.0085 |
| | TCC [15] | 52.53 | 43.85 | 42.86 | 62.33 | 56.08 | **57.89** | 0.1163 | **0.1103** |
| | GTA [22] | 56.92 | 42.97 | 59.96 | 62.79 | 58.52 | 53.32 | -0.2370 | 0.1005 |
| | AE2 (ours) | **66.56** | **57.15** | **65.60** | **65.54** | **65.79** | 57.35 | **0.1380** | 0.0934 |
| (D) | Random Features | 30.31 | 33.42 | 28.10 | 66.47 | 58.98 | 59.87 | -0.0425 | 0.0177 |
| | ImageNet Features | 69.15 | 42.03 | 58.61 | 76.96 | 66.90 | 60.31 | -0.4143 | 0.0734 |
| | single-view TCN [63] | 68.87 | 48.86 | 36.48 | 73.76 | 55.08 | 56.65 | -0.0602 | 0.0737 |
| | CARL [10] | 59.69 | 35.19 | 47.83 | 69.43 | 54.83 | 63.19 | -0.1310 | 0.0542 |
| | TCC [15] | 78.41 | 53.29 | 32.87 | 80.24 | 55.84 | 47.27 | 0.2155 | 0.1040 |
| | GTA [22] | 83.63 | 82.91 | 81.80 | 85.20 | 78.00 | 79.14 | 0.4691 | 0.4901 |
| | AE2 (ours) | **85.87** | **84.71** | **85.68** | **86.83** | **81.46** | **82.07** | **0.5060** | **0.6171** |

or pretrained on ImageNet [14]; (2) Self-supervised learning approaches specifically designed for learning ego-exo view-invariant features, **time-contrastive networks (TCN)** [63] and **ActorObserverNet** [66]. Both require synchronized ego-exo videos for training, and are thus only applicable to Break Eggs data. Additionally, TCN offers a single-view variant that eliminates the need for synchronized data, by taking positive frames within a small temporal window surrounding the anchor, and negative pairs from distant timesteps. We implement **single-view TCN** for all datasets. (3) Fine-grained action representation learning approaches, **CARL** [10] which utilizes a sequence contrastive loss, and **TCC** [15] and **GTA** [22], which also utilize alignment as the pretext task but do not account for ego-exo viewpoint differences.

**Main Results**   In Table 1, we benchmark all baseline approaches and AE2 on four ego-exo datasets. See Supp. for full results, including the F1 score for few-shot classification, and mAP@5,15 for frame retrieval. From the results, it is clear that AE2 greatly outperforms other approaches across all downstream tasks. For instance, on Break Eggs, AE2 surpasses the multi-view TCN [63], which utilizes perfect ego-exo synchronization as a supervision signal, by +6.32% in F1 score. Even when we adapt the TCN to treat all possible ego-exo pairs as perfectly synchronized, AE2 still excels, showing a considerable margin of improvement (see Supp. for a discussion). Pour Liquid poses the most substantial challenge due to its in-the-wild nature, as reflected by the low phase progression and Kendall's Tau values. Yet, AE2 notably improves on previous works, achieving, for instance, a +9.64% F1 score. Regarding Tennis Forehand, another in-the-wild dataset, we note that some methods like TCC perform well in identifying correspondences within either the ego or exo domain, but struggle to connect the two. Consequently, while it achieves an 80.24% mAP@10 for standard frame retrieval, it falls below 60% for cross-view frame retrieval. In contrast, AE2 effectively bridges these two significantly different domains, resulting in a high performance of over 81% mAP@10

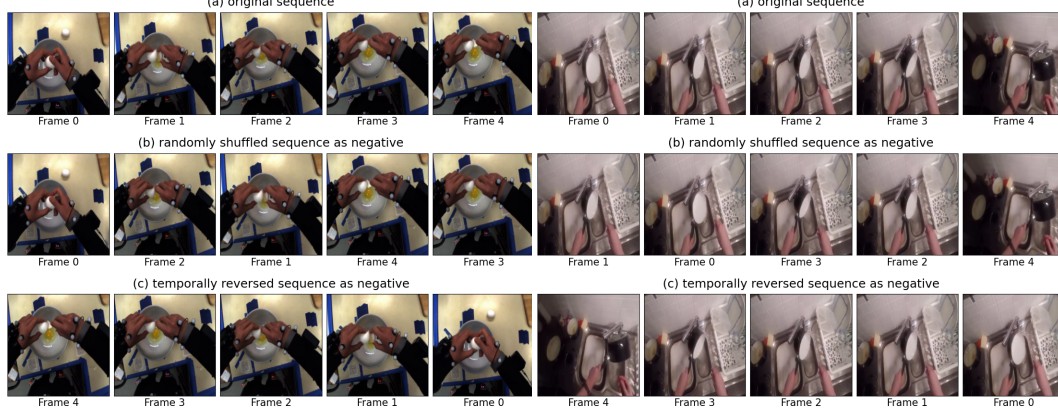

Figure 4: Visualization of (a) the original video sequence; (b) a randomly shuffled sequence; and (c) a temporally reversed sequence on Break Eggs (left) and Pour Liquid (right). The randomly shuffled sequence may still preserve the action progression order due to many similar frames in the video sequence, failing to be truly "negative". In contrast, temporally reversing frames results in a more distinct and suitable negative example.

in both standard and cross-view settings. These results demonstrate the efficacy of AE2 in learning fine-grained ego-exo view-invariant features.

**Ablation**  Table 2 provides an ablation of the two design choices of AE2. Starting with a baseline approach (denoted as "Base DTW" in the table), we optimize a ResNet-50 encoder with respect to a DTW alignment objective as defined in Eqn. (3). Next, we replace the feature encoder with our proposed object-centric encoder design and report its performance. Finally, we incorporate the contrastive regularization from Eqn. (4) into the objective. It is evident that object-centric representations are instrumental in bridging the ego-exo gap, leading to substantial performance improvements. Furthermore, incorporating contrastive regularization provides additional gains.

Table 2: Ablation study of AE2 on four datasets (F1 score). See Supp. for other metrics.

|  | Dataset | | | |
|---|---|---|---|---|
|  | (A) | (B) | (C) | (D) |
| Base DTW | 58.53 | 82.91 | 59.66 | 79.56 |
| + object | 62.86 | 84.04 | 63.28 | 84.14 |
| + object + contrast | **66.23** | **85.17** | **66.56** | **85.87** |

**Negative Sampling**  We conduct experiments to validate the efficacy of creating negative samples by temporally reversing frames, as opposed to random shuffling and report results in Table 3. In general, temporally reversing frames yields superior and more consistent performance than randomly shuffling. The inferior performance of random shuffling can be related to the abundance of similar frames within the video sequence. Even after shuffling, the frame sequence may still emulate the natural progression of an action, thereby appearing more akin to a positive sample. Conversely, unless the frame sequence is strictly symmetric (a scenario that is unlikely to occur in real

Table 3: Comparison of AE2 employing randomly shuffled frames versus temporally reversed frames as the negative samples (Frame Retrieval mAP@10). See Supp. for other metrics.

|  | Dataset | | | |
|---|---|---|---|---|
|  | (A) | (B) | (C) | (D) |
| Random Shuffle | 61.66 | 81.92 | 63.48 | 86.73 |
| Temporal Reverse | **65.85** | **84.90** | **65.54** | **86.83** |

videos), temporally reversing frames is apt to create a negative sample that deviates from the correct action progression order. To further illustrate this point, we visualize two video sequences in Fig. 5. It is evident that the randomly shuffled sequence seems to preserve the sequence of actions like breaking eggs or pouring liquid, thereby resembling a "positive" example. Consequently, incorporating such negative samples into training may confuse the model and lead to diminished performance.

**Qualitative Results**  Fig. 5 (left) shows examples of cross-view frame retrieval on Pour Liquid and Tennis Forehand datasets. Given a query image from one viewpoint in the test set, we retrieve its nearest neighbor among all test frames in another viewpoint. We observe that the retrieved frames closely mirror the specific action state of the query frame. For example, when the query is a frame

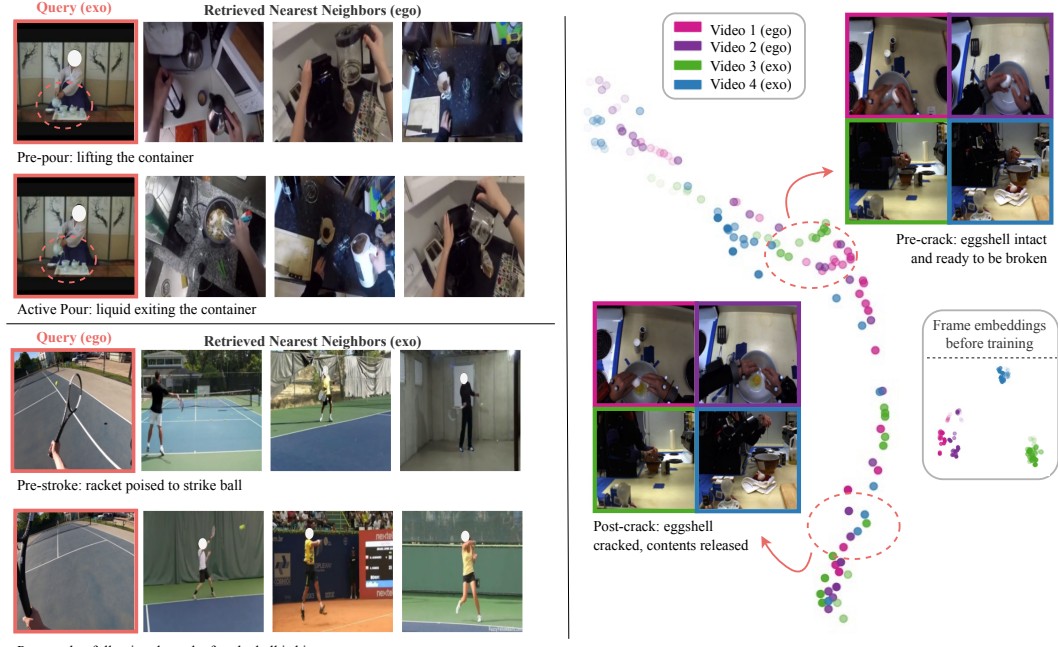

Figure 5: Qualitative results of AE2. Left: Cross-view frame retrieval results on Pour Liquid and Tennis Forehand. Our view-invariant embeddings can be effectively used for fine-grained frame retrieval; Right: tSNE trajectories of 4 test videos (2 ego and 2 exo) in the embedding space on Break Eggs. Shading indicates time. The learned embeddings successfully capture the progress of an action and maintain invariance across the ego and exo viewpoints. See Supp. for more examples.

capturing the moment before the pouring action, where the liquid is still within the container (row 1), or when the query shows pouring actively happening (row 2), the retrieved exo frames capture that exact step as well. Fig. 5 (right) shows tSNE visualizations of embeddings for 4 test videos on Break Eggs. Our learned embeddings effectively capture the progress of an action, while remaining view-invariant. These learned embeddings enable us to transfer the pace of one video to others depicting the same action. We demonstrate this capability with examples of both ego and exo videos, collected in distinct environments, playing synchronously in the Supp. video.

**Limitations** While AE2 has demonstrated its ability to handle in-the-wild data, there remains much room for improvement, particularly on challenging in-the-wild datasets such as Pour Liquid. Here, we observe instances of failure in cross-view frame retrieval and synchronizing the pace of two videos. Although we highlight successful examples in our discussion, it is important to note that some attempts have been unsuccessful, largely due to the inherent noise in real-world videos. This represents a key area for future refinement and development of AE2.

## 6 Conclusion

We tackle fine-grained temporal understanding by jointly learning from *unpaired* egocentric and exocentric videos. Our core idea is to leverage temporal alignment as the self-supervised learning objective for invariant features, together with a novel object-centric encoder and contrastive regularizer that reverses frames. On four datasets we achieve superior performance compared to state-of-the-art ego-exo and alignment approaches. This fundamental technology has great promise for both robot learning and human skill learning in AR, where an ego actor needs to understand (or even match) the actions of a demonstrator that they observe from the exo point of view. Future work includes generalizing AE2 to train from pooled data of multiple actions, extending its applicability to more complex, longer action sequences and exploring its impact for robot imitation learning.

## Acknowledgements

UT Austin is supported in part by the IFML NSF AI Institute. KG is paid as a research scientist at Meta. The authors extend heartfelt thanks to the following individuals from UT Austin for their vital role in collecting the ego-tennis forehand dataset: Bowen Chen, Zhenyu Jiang, Hanwen Jiang, Xixi Hu, Ashutosh Kumar, Huihan Liu, Bo Liu, Mi Luo, Sagnik Majumder, Shuhan Tan, and Yifeng Zhu. We are also grateful to Zhengqi Gao (MIT), Xixi Hu (UT Austin) and Yue Zhao (UT Austin), for their constructive discussions and feedback.

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

# A   Video Containing Qualitative Results

We invite the reader to view the video available at https://vision.cs.utexas.edu/projects/AlignEgoExo/, where we show qualitative examples of one practical application enabled by our learned ego-exo view-invariant representations—synchronous playback of egocentric and exocentric videos. We randomly select ego-exo video pairs from the test set, and use the frozen encoder $\phi$ to extract frame-wise embeddings. We then match each frame of one video (the reference) to its closest counterpart in the other video using nearest neighbor. As demonstrated across all datasets with several examples, AE2 effectively aligns two videos depicting the same action, overcoming substantial viewpoint and background differences. The video also includes examples of synchronizing two egocentric or two exocentric videos.

# B   Experimental Setup

## B.1   Datasets

Since existing video datasets for fine-grained video understanding (*e.g.*, PennAction [82], Fine-Gym [65], IKEA ASM [3]) are solely composed of third-person videos, we curate video clips from five public datasets: Break Eggs [13], H2O [32], EPIC-Kitchens [11], HMDB51 [31] and Penn Action [82] and collect a egocentric Tennis Forehand dataset. Our data selection criteria is to find videos that depict the same action from distinct egocentric and exocentric viewpoints. Consequently, TCN pouring [63] is excluded due to its small scale and significant similarity between the egocentric and exocentric views; Assembly101 [62] is not selected since there there are large variations within a single atomic action and the egocentric video is monochromatic.

In all, our dataset compilation from five public data sources, along with our collected egocentric tennis videos, results in four distinct ego-exo datasets, each describing specific atomic actions:

(A) **Break Eggs**. The dataset contains 44 subjects cooking five different recipes (brownies, pizza, sandwich, salad, and scrambled eggs), captured from one egocentric and four exocentric views simultaneously in CMU-MMAC [13]. We use the temporal keystep boundaries provided in [2] to extract clips corresponding to the action of breaking eggs from all videos. Among the four exocentric viewpoints, we adopt videos from the right-handed view, as this viewpoint captures the action being performed most clearly. We randomly split the data into training and validation sets across subjects, with 35 subjects (118 videos) for training and 9 subjects (30 videos) for validation and test. There is strict synchronization between egocentric and exocentric video pairs in this dataset.

(B) **Pour Milk**. The dataset features 10 subjects interacting with a milk carton using both hands, captured by one egocentric camera and four static exocentric cameras in H2O [32]. We utilize the keystep annotations provided in [33] to extract clips corresponding to the pouring milk action. All four exocentric views are included, as they clearly capture the action. We follow the data split in [33], with 7 subjects (77 videos) in the training set and 3 subjects (35 videos) in the validation and test set. A portion of this dataset (the first 3 subjects) contains synchronized egocentric-exocentric video pairs, while the remaining part does not.

(C) **Pour Liquid**. To evaluate our methods on in-the-wild data, we assemble a Pour Liquid dataset by extracting clips from one egocentric dataset, EPIC-Kitchens [11] and one exocentric dataset, HMDB51 [31]. We utilize clips from the "pour water" class in EPIC-Kitchens and "pour" category in HMDB51. Following the data split in the original datasets, we obtain 137 videos for training and 56 videos for validation and test. It is important to note that

Table 4: Dataset summary.

| Dataset | # Train | | # Val | | # Test | | Fixed exo view? | Ego-exo time-sync? |
|---|---|---|---|---|---|---|---|---|
| | ego | exo | ego | exo | ego | exo | | |
| (A) Break Eggs | 61 | 57 | 5 | 5 | 10 | 10 | ✓ | ✓ |
| (B) Pour Milk | 29 | 48 | 4 | 8 | 7 | 16 | ✓ | ✗ |
| (C) Pour Liquid | 70 | 67 | 10 | 9 | 19 | 18 | ✗ | ✗ |
| (D) Tennis Forehand | 94 | 79 | 25 | 24 | 50 | 50 | ✗ | ✗ |

the egocentric and exocentric videos are neither synchronized nor collected in the same environment, providing a challenging testbed.

(D) **Tennis Forehand**. To include physical activities in our study, we leverage exocentric video sequences of the tennis forehand action from Penn Action [82] and collect an egocentric dataset featuring the same action performed by 12 subjects using Go Pro HERO8. We adopt the data split from [15] for Penn Action, and divide our egocentric tennis forehand dataset by subject: 8 for training and 4 for validation and testing. This results in a total of 173 clips for training, and 149 clips for validation and testing. The egocentric and exocentric videos, gathered from a range of real-world scenarios, are naturally unpaired.

Table 4 provides a summary of these four datasets. In addition, we recognize that the original datasets lack frame-wise labels and provide dense frame-level annotations to enable a comprehensive evaluation of the learned representations. See Table 5 for a complete list of all the key events we annotate and Fig. 6 for illustrative examples. We will release our collected data and labels for academic usage. [5]

Table 5: Number of actions phases and list of key events for each dataset.

| Dataset | # phases | List of key events |
|---|---|---|
| (A) Break Eggs | 4 | hit egg, visible crack on the eggshell; egg contents released into bowl |
| (B) Pour Milk | 3 | liquid starts exiting, pouring complete |
| (C) Pour Liquid | 3 | liquid starts exiting, pouring complete |
| (D) Tennis Forehand | 2 | racket touches ball |

## B.2 Evaluation

We provide a detailed description of the four downstream tasks below and their corresponding evaluation metrics:

1. **Action Phase Classification**. We train an SVM classifier on top of the embeddings to predict the action phase labels for each frame and report F1 score on test data. Besides the regular setting, we investigate (1) few-shot; and (2) cross-view zero-shot settings.

   (1) Few-shot. We assume that only a limited number of training videos have annotations and can be used for training the SVM classifier.

   (2) Cross-view zero-shot. We assume that per-frame labels of training data are only available on one view for training the SVM classifier, and report the test performance on the other view. We use the terms "exo2ego" to describe the case where we use exocentric data for training the SVM classifier and test its performance on egocentric data, while "ego2exo" represents the reverse case.

2. **Phase Progression**. We train a linear regressor on the frozen embeddings to predict the phase progression values, defined as the difference in time-stamps between any given frame and each key event, normalized by the number of frames in that video [15]. Average R-square measure on test data is reported. This metric evaluates how well the progress of an action is captured by the embeddings, with the maximum value being 1.

3. **Frame Retrieval**. We report the mean average precision (mAP)@K (K=5,10 ,15). For each query, average precision is computed by determining how many frames among the retrieved K frames have the same action phase labels as the query frame, divided by K. Furthermore, to evaluate view-invariance, we propose the cross-view frame retrieval task (*i.e.*, ego2exo and exo2ego frame retrieval). For each query in one view, the goal is to retrieve K frames from another view. No additional training is required for this task.

4. **Kendall's Tau**. This metric is calculated for every pair of test videos by sampling two frames in the first video and retrieving the corresponding nearest frames in the second video, then checking whether their orders are shuffled. It measures how well-aligned two sequences are in time. No additional training or frame-wise labels are necessary for this evaluation.

---

[5]This dataset is owned by UT Austin and involves no participation from Meta.

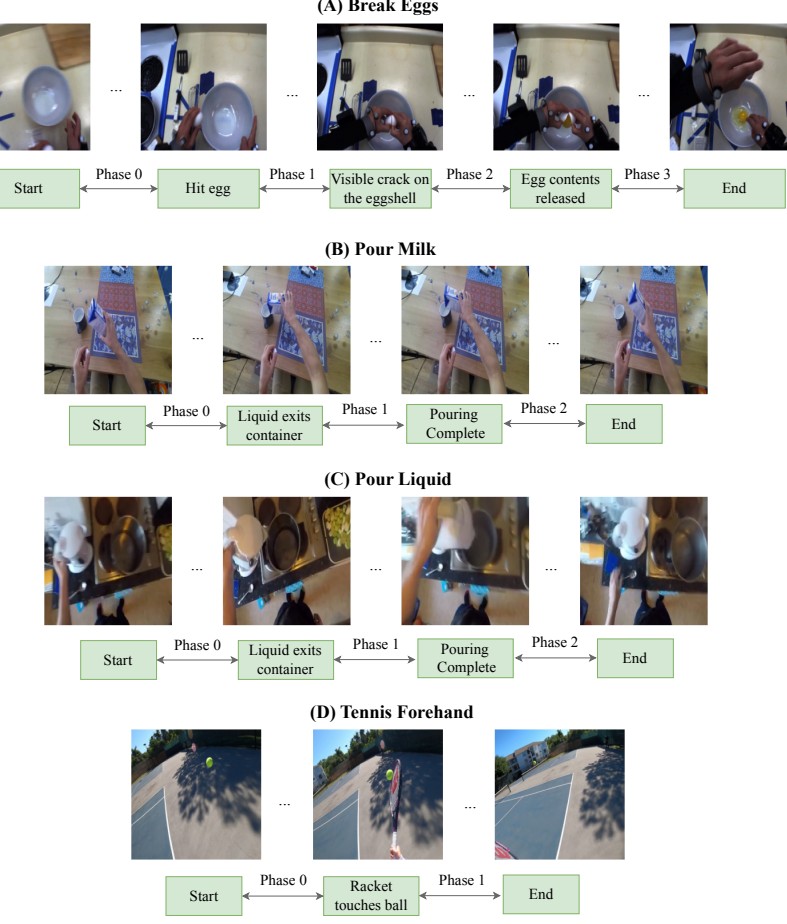

Figure 6: Example labels for all datasets. Key events are displayed in boxes below sequences, with the phase label assigned to each frame between two key events.

Table 6: Hyperparameters summary. 'Lr' stands for learning rate, and 'Wd' denotes weight decay.

| Datasets | Optimizer | | Transformer Encoder | | Regularization | | |
| | Lr | Wd | Hidden Dim. | # Layers | # Frames | # Pos. Frames | Ratio $\lambda$ |
| --- | --- | --- | --- | --- | --- | --- | --- |
| (A) Break Eggs | 5e-5 | 1e-5 | 256 | 1 | 32 | 32 | 1 |
| (B) Pour Milk | 1e-4 | 1e-5 | 256 | 1 | 32 | 8 | 2 |
| (C) Pour Liquid | 5e-5 | 1e-5 | 128 | 3 | 32 | 16 | 2 |
| (D) Tennis Forehand | 5e-5 | 1e-5 | 128 | 1 | 20 | 10 | 4 |

## B.3 Implementation

For all video sequences, frames are resized to $224 \times 224$. During training, we randomly extract 32 frames from each video to construct a video sequence. We train the models for a total number of 300 epochs with a batch size of 4, using the Adam optimizer. The base encoder $\phi_{\text{base}}$ is initialized with a ResNet-50 pretrained on ImageNet, and jointly optimized with the transformer encoder $\phi_{\text{transformer}}$ throughout the training process. The model checkpoint demonstrating the best performance on validation data is selected, and its performance on test data is reported. In terms of the encoder network, global features are taken from the output of $Conv4c$ layer in $\phi_{\text{base}}$. Following [15], we stack the features of any given frame and its context frames along the dimension of time, followed by 3D convolutions for aggregating temporal information and 3D max pooling. In all experiments, the number and stride of context frames are set as 1 and 15, respectively. For local features, we take output of the $Conv1$ layer in $\phi_{\text{base}}$ and apply 3D max pooling to aggregate temporal information from the given frame and its context frame. The features are then fed as input of ROI Align.

Table 7: Results of few-shot action phase classification (F1 score) and frame retrieval (mAP@5,10,15).

| Dataset | Method | Few-shot Cls. | | | Frame Retrieval | | |
|---|---|---|---|---|---|---|---|
| | | 10% | 50% | 100% | mAP@5 | mAP@10 | mAP@15 |
| (A) | Random Features | 19.18 | 19.18 | 19.18 | 48.26 | 47.13 | 45.75 |
| | ImageNet Features | 46.15 | 48.80 | 50.24 | 49.98 | 50.49 | 50.08 |
| | ActorObserverNet [66] | 31.40 | 35.63 | 36.14 | 50.92 | 50.47 | 49.72 |
| | single-view TCN [63] | 52.30 | 54.90 | 56.90 | 52.82 | 53.42 | 53.60 |
| | multi-view TCN [63] | 56.88 | _59.25_ | _59.91_ | 59.11 | 58.83 | 58.44 |
| | multi-view TCN (unpaired) [63] | 56.13 | 56.65 | 56.79 | 58.18 | 57.78 | 57.21 |
| | CARL [10] | 39.18 | 41.92 | 43.43 | 47.14 | 46.04 | 44.99 |
| | TCC [15] | _57.54_ | 59.18 | 59.84 | 59.33 | 58.75 | 57.99 |
| | GTA [22] | 56.89 | 56.77 | 56.86 | _62.79_ | _61.55_ | _60.38_ |
| | AE2 (ours) | **63.95** | **64.86** | **66.23** | **66.86** | **65.85** | **64.73** |
| (B) | Random Features | 36.84 | 36.84 | 36.84 | 52.94 | 52.48 | 51.59 |
| | ImageNet Features | 39.29 | 40.83 | 41.59 | 53.32 | 54.09 | 54.06 |
| | single-view TCN [63] | 43.60 | 46.83 | 47.39 | 56.98 | 57.00 | 56.46 |
| | CARL [10] | 48.73 | 48.78 | 48.79 | 55.59 | 55.01 | 54.23 |
| | TCC [15] | 78.69 | 77.97 | 77.91 | _81.22_ | _80.97_ | _80.46_ |
| | GTA [22] | _79.82_ | _80.96_ | _81.11_ | 80.65 | 80.12 | 79.68 |
| | AE2 (ours) | **85.17** | **85.12** | **85.17** | **85.25** | **84.90** | **84.55** |
| (C) | Random Features | 45.26 | 45.26 | 45.26 | 49.69 | 49.83 | 49.18 |
| | ImageNet Features | 55.53 | 54.43 | 53.13 | 50.52 | 51.49 | 51.89 |
| | single-view TCN [63] | 54.62 | 55.08 | 54.02 | 48.50 | 48.83 | 49.03 |
| | CARL [10] | 51.68 | 55.67 | _56.98_ | 55.03 | 55.29 | 54.93 |
| | TCC [15] | 52.37 | 51.70 | 52.53 | _62.93_ | 62.33 | 61.44 |
| | GTA [22] | _55.91_ | _56.87_ | 56.92 | 62.83 | _62.79_ | _62.12_ |
| | AE2 (ours) | **65.88** | **66.53** | **66.56** | **66.55** | **65.54** | **64.66** |
| (D) | Random Features | 31.54 | 30.31 | 30.31 | 69.57 | 66.47 | 64.34 |
| | ImageNet Features | 65.48 | 68.03 | 69.15 | 78.11 | 76.96 | 75.84 |
| | single-view TCN [63] | 65.78 | 69.19 | 68.87 | 74.05 | 73.76 | 73.10 |
| | CARL [10] | 58.89 | 59.38 | 59.69 | 72.94 | 69.43 | 67.14 |
| | TCC [15] | 67.71 | 77.07 | 78.41 | 82.78 | 80.24 | 78.59 |
| | GTA [22] | _80.31_ | _83.04_ | _83.63_ | _86.59_ | _85.20_ | _84.33_ |
| | AE2 (ours) | **85.24** | **85.72** | **85.87** | **87.94** | **86.83** | **86.05** |

**During evaluation, we freeze the encoder $\phi$ and use it to extract 128-dimensional embeddings for each frame.** These representations are then assessed across a variety of downstream tasks (Sec. 4). Detailed hyperparameters specific to each dataset are provided in Table 6. Noteworthy adjustments include: (1) In the case of Tennis Forehand, we utilize a single object proposal, as the active object is only the tennis racket (the tennis ball is too small to be detected reliably). Furthermore, given the shorter video lengths, we sample 20 frames from each video as opposed to the usual 32. (2) For datasets featuring non-monotonic actions (*i.e.*, Pour Milk and Pour Liquid), we construct the negative sequence by randomly reversing either the first or the last half of the sequence, rather than the whole sequence. This is due to the cyclic nature of the pouring action present in some videos within these datasets. All experiments are conducted using PyTorch [52] on 2 Nvidia V100 GPUs.

## C  Further Results and Visualizations

**Results**  Supplementing Table 1 in the main paper, Tables 7 and 8 present comprehensive results of AE2 and baseline models on few-shot action phase classification and frame retrieval tasks. For few-shot classification, we train the SVM classifier with 10% (or 50%) of the training data, averaging results over 10 runs. AE2 demonstrates superior performance in learning fine-grained, view-invariant ego-exo features when compared with ego-exo [66, 63], frame-wise contrastive learning [10], and alignment-based [15, 22] approaches.

On Break Eggs, AE2 greatly outperforms the multi-view TCN [63], which utilizes perfect ego-exo synchronization as a supervision signal. We hypothesize that the strict supervision requirement of

Table 8: Results of cross-view frame retrieval (mAP@5,10,15).

| Dataset | Method | Ego2exo Frame Retrieval | | | Exo2ego Frame Retrieval | | |
|---|---|---|---|---|---|---|---|
| | | mAP@5 | mAP@10 | mAP@15 | mAP@5 | mAP@10 | mAP@15 |
| (A) | Random Features | 42.51 | 41.74 | 40.51 | 38.08 | 38.19 | 37.10 |
| | ImageNet Features | 33.32 | 33.09 | 32.78 | 38.99 | 37.80 | 36.71 |
| | ActorObserverNet [66] | 43.57 | 42.70 | 41.56 | 42.00 | 41.29 | 40.48 |
| | single-view TCN [63] | 31.12 | 32.63 | 33.73 | 34.67 | 34.91 | 35.31 |
| | multi-view TCN [63] | 46.38 | 47.04 | 46.96 | 52.50 | 52.68 | 52.43 |
| | multi-view TCN (unpaired) [63] | 55.34 | 54.64 | 53.75 | 58.79 | 57.87 | 57.07 |
| | CARL [10] | 37.89 | 37.38 | 36.57 | 40.37 | 39.94 | 39.38 |
| | TCC [15] | 62.11 | 61.11 | 60.33 | 62.39 | 62.03 | 61.25 |
| | GTA [22] | 57.11 | 56.25 | 55.10 | 54.47 | 53.93 | 53.22 |
| | AE2 (ours) | **65.70** | **64.59** | **63.76** | **62.48** | **62.15** | **61.80** |
| (B) | Random Features | 51.46 | 50.56 | 48.93 | 52.78 | 51.98 | 50.82 |
| | ImageNet Features | 25.72 | 27.31 | 28.57 | 41.50 | 43.21 | 43.06 |
| | single-view TCN [63] | 47.00 | 46.48 | 45.42 | 47.94 | 47.20 | 46.59 |
| | CARL [10] | 54.35 | 52.99 | 51.99 | 51.14 | 51.51 | 51.00 |
| | TCC [15] | 75.54 | 75.30 | 75.02 | 80.44 | 80.27 | 80.18 |
| | GTA [22] | 72.55 | 72.78 | 72.96 | 75.16 | 75.40 | 75.48 |
| | AE2 (ours) | **78.21** | **78.48** | **78.78** | **83.88** | **83.41** | **83.05** |
| (C) | Random Features | 55.78 | 55.44 | 54.77 | 56.31 | 55.75 | 54.56 |
| | ImageNet Features | 51.44 | 52.17 | 52.38 | 30.18 | 30.44 | 30.40 |
| | single-view TCN [63] | 53.60 | 55.28 | 55.46 | 29.16 | 31.15 | 31.95 |
| | CARL [10] | 59.59 | 59.37 | 59.19 | 34.73 | 36.80 | 38.10 |
| | TCC [15] | 55.98 | 56.08 | 56.13 | **58.11** | **57.89** | **57.15** |
| | GTA [22] | 57.03 | 58.52 | 59.00 | 51.71 | 53.32 | 53.54 |
| | AE2 (ours) | **66.23** | **65.79** | **65.00** | 57.42 | 57.35 | 57.03 |
| (D) | Random Features | 61.24 | 58.98 | 56.94 | 63.42 | 59.87 | 57.57 |
| | ImageNet Features | 69.34 | 66.90 | 64.95 | 61.61 | 60.31 | 58.55 |
| | single-view TCN [63] | 54.12 | 55.08 | 55.05 | 56.70 | 56.65 | 55.84 |
| | CARL [10] | 52.18 | 54.83 | 55.39 | 65.94 | 63.19 | 60.83 |
| | TCC [15] | 57.87 | 55.84 | 53.81 | 48.62 | 47.27 | 46.11 |
| | GTA [22] | 78.93 | 78.00 | 77.01 | 79.95 | 79.14 | 78.52 |
| | AE2 (ours) | **82.58** | **81.46** | **80.75** | **82.82** | **82.07** | **81.69** |

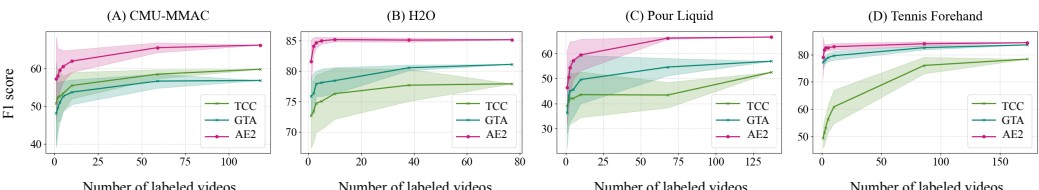

Figure 7: Few-shot action phase classification results. AE2 achieves superior performance across a wide range of labeled training videos, particularly under the most challenging conditions where less than 10 videos are labeled. Results are averaged over 50 runs, and confidence bars represent one standard deviation.

TCN might be limiting, as it can not utilize as many ego-exo pairs as AE2 due to its reliance on ego-exo synchronization. In contrast, AE2 capitalizes on a broader set of unpaired ego-exo data. Even when we modify multi-view TCN to consider all potential ego-exo pairs as synchronously perfect (termed as multi-view TCN (unpaired) in the tables), it does not outperform its regular version, indicating a lack of robustness towards non-synchronized ego-exo pairs. Consequently, it appears that multi-view TCN is ill-equipped to learn desired view-invariant representations from unpaired, real-world ego-exo videos.

Table 9: Ablation Study of AE2.

| Dataset | Method | Classification (F1 score) | | | Frame Retrieval (mAP@10) | | | Phase Progression | Kendall's Tau |
|---|---|---|---|---|---|---|---|---|---|
| | | regular | ego2exo | exo2ego | regular | ego2exo | exo2ego | | |
| (A) | Base DTW | 58.53 | _57.78_ | 54.23 | 58.36 | 55.36 | 58.95 | 0.1920 | _0.5641_ |
| | + object | _62.86_ | **60.88** | _58.52_ | _62.66_ | _61.26_ | _60.69_ | _0.4235_ | 0.5484 |
| | + object + contrast | **66.23** | 57.41 | **71.72** | **65.85** | **64.59** | **62.15** | **0.5109** | **0.6316** |
| (B) | Base DTW | 82.91 | 81.82 | 81.83 | 81.49 | 74.63 | 80.21 | 0.7525 | 0.8199 |
| | + object | _84.04_ | _84.20_ | **84.23** | _83.03_ | **81.42** | _81.57_ | **0.7646** | _0.8886_ |
| | + object + contrast | **85.17** | **84.73** | _82.77_ | **84.90** | _78.48_ | **83.41** | _0.7634_ | **0.9062** |
| (C) | Base DTW | 59.66 | 55.48 | 59.49 | 52.57 | 54.12 | 52.23 | 0.0553 | 0.0609 |
| | + object | _63.28_ | **57.60** | _62.42_ | _63.40_ | **67.15** | **63.05** | **0.2231** | **0.1339** |
| | + object + contrast | **66.56** | _57.15_ | **65.60** | **65.54** | _65.79_ | _57.35_ | _0.1380_ | _0.0934_ |
| (D) | Base DTW | 79.56 | 81.38 | 72.54 | 82.65 | 75.36 | 76.74 | 0.4022 | 0.4312 |
| | + object | _84.14_ | **85.36** | _83.32_ | **88.22** | _79.07_ | **82.61** | **0.5431** | **0.6477** |
| | + object + contrast | **85.87** | _84.71_ | **85.68** | _86.83_ | **81.46** | _82.07_ | _0.5060_ | _0.6171_ |

**Few-shot Learning**    Besides the few-shot results in Table 7, we vary the number of labeled training videos, ranging from extremely sparse (a single labeled video) to the case where all training videos are labeled. Note that each labeled video equates to multiple labeled frames. AE2 is compared with top-performing baselines, TCC [15] and GTA [22], across all four datasets in Fig. 7. The results are averaged over 50 runs and include a +- one standard deviation error bar. As shown, AE2 excels in low-label scenarios. For instance, on Pour Milk, a single labeled video yields an action phase classification F1 score over 80%. This suggests that AE2 effectively aligns representations across all training videos, enabling a robust SVM classifier for the downstream task, even with minimal labeling.

**Ablation**    Table 9 presents an ablation of AE2 on all four datasets, which is a comprehensive version of Table 2 in the main paper. From the results, we can see that object-centric representations are instrumental in bridging the ego-exo gap, leading to substantial performance improvements. For instance, frame retrieval mAP@10 improves by +10.83% on Pour Liquid and +5.57% on Tennis Forehand. Furthermore, incorporating contrastive regularization provides additional performance boosts for several downstream tasks such as regular action phase classification. These results demonstrate the integral contributions of both components of AE2 to achieve optimal performance.

**Negative Sampling (Temporal Reverse versus Random Shuffle)**    Table 10 presents a comparison of two negative sampling strategies (random shuffling versus temporal reversing), which is a comprehensive version of Table 3 in the main paper. The results reveal that, in general, temporally reversing frames yields superior and more consistent performance than randomly shuffling. For example, on Break Eggs data, random shuffling results in a decrease of -5.62% in F1 score and -4.19% in mAP@10 compared with regular AE2. The inferior performance of random shuffling can be related to the abundance of similar frames within the video sequence. Even after shuffling, the frame sequence may still emulate the natural progression of an action, thereby appearing more akin to a positive sample. Conversely, unless the frame sequence is strictly symmetric (a scenario that is unlikely to occur in real videos), temporally reversing frames is apt to create a negative sample that deviates from the correct action progression order.

To dissect the performance gains further, the large improvement observed on dataset (A) when using temporally reversed frames over randomly shuffled frames stems from its inherent ability to be more robust to the repetitive nature of the breaking egg action (e.g., the camera wearer may hit the egg twice before breaking it, resulting in many similar frames in the video sequence). On the other hand, the subtler improvement on dataset (D), dealing with the tennis forehand action, is likely due to the strictly monotonic nature of this action, leaving less room for our method to outperform. However, it is essential to emphasize that this does not diminish the value of our approach, our negative sampling strategy still proves preferable consistently across all four datasets.

**Negative Sampling (Sequence-level versus Frame-level)**    We emphasize the unique advantages of our proposed sequence-level negatives over the frame-level negatives used in prior works [63, 66, 10].

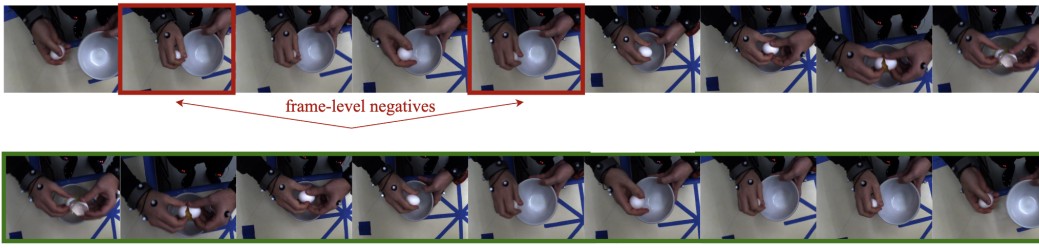

Temporally away but visually similar frames

frame-level negatives

sequence-level negatives

Figure 8: Visualization of a video sequence (from CMU-MMAC) exhibiting a certain amount of periodicity, where the camera wearer hits the egg to the bowl's edge twice. In the upper figure, frame-level negatives are defined based on temporal distance, leading to the possibility of two visually similar frames (marked by a red border) being misclassified as a "negative" pair. In contrast, the lower figure illustrates how sequence-level negatives provide a more robust solution to accomodate the temporal variations within a video.

Table 10: Comparison of AE2 employing randomly shuffled frames versus temporally reversed frames as the negative samples.

| Dataset | Method | Classification (F1 score) | | | Frame Retrieval (mAP@10) | | | Phase |
| | | regular | ego2exo | exo2ego | regular | ego2exo | exo2ego | Progression |
| --- | --- | --- | --- | --- | --- | --- | --- | --- |
| (A) | Random Shuffle | 60.61 | **58.63** | 54.46 | 61.66 | 57.57 | 59.61 | 0.3385 |
| | Temporal Reverse | **66.23** | 57.41 | **71.72** | **65.85** | **64.59** | **62.15** | **0.5109** |
| (B) | Random Shuffle | 82.05 | 82.74 | 79.48 | 81.92 | **80.60** | 79.71 | 0.6659 |
| | Temporal Reverse | **85.17** | **84.73** | **82.77** | **84.90** | 78.48 | **83.41** | **0.7634** |
| (C) | Random Shuffle | **68.13** | **64.96** | 58.44 | 63.48 | 62.73 | **60.27** | -0.0121 |
| | Temporal Reverse | 66.56 | 57.15 | **65.60** | **65.54** | **65.79** | 57.35 | **0.1380** |
| (D) | Random Shuffle | 84.12 | **84.83** | 82.59 | 86.73 | 80.49 | **83.75** | **0.5304** |
| | Temporal Reverse | **85.87** | 84.71 | **85.68** | **86.83** | **81.46** | 82.07 | 0.5060 |

While periodic actions present great challenges for all negative sampling techniques, sequence-level negatives offers a more robust solution to tackle the intricacies of repetitiveness within an action. As a motivating example, Fig. 8 depicts a video sequence of breaking eggs with a certain amount of periodicity (where the camera wearer hits the egg to the bowl's edge twice). Frame-level sampling approaches treat temporally close frames as positive pairs and temporally distant ones as negative pairs. As a consequence, the two visually similar frames (marked by a red border) would be incorrectly identified as a negative pair by those existing methods. Contrarily, our method of constructing negative samples adopts a global perspective, addressing temporal fluctuations within a video more effectively. By reversing the entire sequence, we create a challenge in aligning the reversed ego view video with the original exo view video, thus forming a valid negative sample.

This methodology is not only conceptually appealing but also empirically validated. Table 1 in the main paper illustrates the effectiveness of our proposed AE2, demonstrating that it outperforms prior techniques utilizing frame-wise negatives [63, 66, 10]. These results offer compelling empirical evidence that our sequence-level negative sampling strategy successfully mitigates the issues related to periodicity and improves performance, affirming its superiority over conventional frame-by-frame sampling methods.

**Visualizations** In addition to the cross-view frame retrieval results for Pour Liquid and Tennis Forehand presented in the main paper (Fig. 5), we showcase results for the other two datasets (*i.e.*, Break Eggs and Pour Milk) in Fig. 9. For any given query frame from one view, the retrieved nearest neighbors closely match the action stage of the query, regardless of substantial differences in viewpoints. These results underline AE2's efficacy in learning fine-grained action representations that transcend ego-exo viewpoint differences.

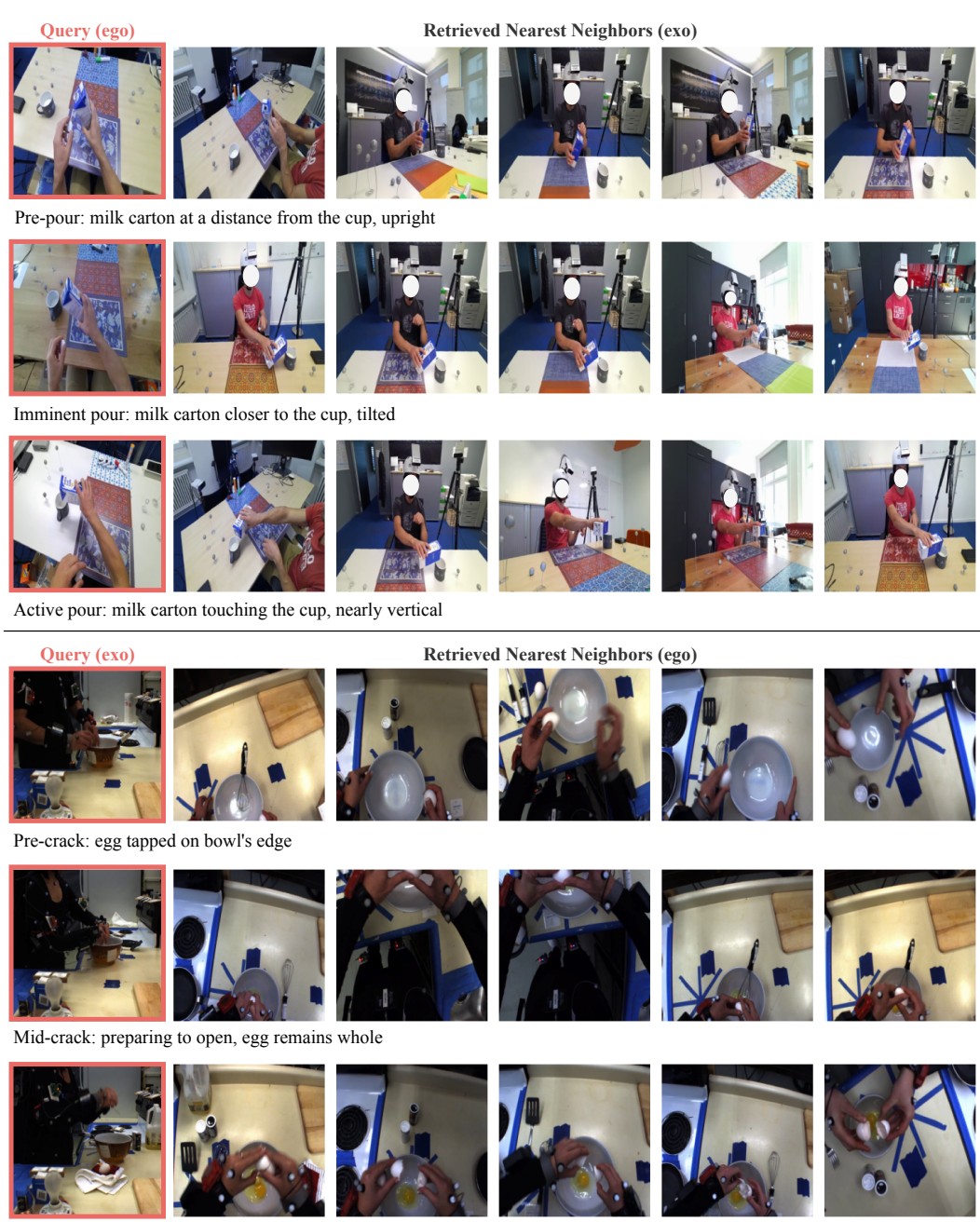

**Query (ego)**      **Retrieved Nearest Neighbors (exo)**

Pre-pour: milk carton at a distance from the cup, upright

Imminent pour: milk carton closer to the cup, tilted

Active pour: milk carton touching the cup, nearly vertical

**Query (exo)**      **Retrieved Nearest Neighbors (ego)**

Pre-crack: egg tapped on bowl's edge

Mid-crack: preparing to open, egg remains whole

Post-crack: eggshell cracked, contents released

Figure 9: Cross-view frame retrieval results on Pour Milk (rows 1-3) and Break Eggs (rows 4-6). AE2 leads to representations that encapsulate the fine-grained state of an action and are invariant to the ego-exo viewpoints.

