# OpenReview forum: "Learning Fine-grained View-Invariant Representations from Unpaired Ego-Exo Videos via Temporal Alignment"
_NeurIPS.cc/2023/Conference — NeurIPS 2023 poster_

### Official Review · Reviewer_G2AN · 2023-07-05

**Soundness:** 2 fair
**Presentation:** 3 good
**Contribution:** 2 fair
**Rating:** 5
**Confidence:** 4

**Summary:**

This paper proposes a self-supervised approach for learning view-invariant action features by aligning the egocentric and exocentric videos in time. The proposed method has two main components: 1)  an object-centric encoder to learn discriminative features around the hands and active objects; 2) a contrastive learning paradigm to align videos from different viewpoints.

Apart from the proposed method, this paper also establishes a new benchmark to evaluate the proposed method, and extensive experiments have been conducted to validate the effectiveness of the proposed method.

**Strengths:**

1. The paper is well-motivated and well-written. Learning view-invariant features for fine-grained action recognition is important in video understanding.

2. Experiments are conducted on four different downstream tasks to show the effectiveness of the proposed method.

**Weaknesses:**

1. The proposed method has limited novelty, and it is a combination of different existing modules. For example, emphasizing the hand-region features and contrastive learning with inverse temporal orders are widely used in many existing works.

2. The proposed benchmark contains four datasets, but each dataset has a very limited number of samples, conversing a few action classes.
(a) Limited samples: e.g., the CMU-MMAC dataset contains 118 videos for training and 30 videos for validation and testing. This makes the model training easy to overfit.
(b) Limited diversity: e.g., the CMU-MMAC dataset contains only a few actions related to 'breaking eggs', which casts doubt on the generalizability of the proposed method.

**Questions:**

As the authors generate the benchmark datasets on their own, please elaborate more on the details, such as the number of action classes in each dataset.

**Limitations:**

Please see the weakness.

---

> ### Author Rebuttal · Authors · 2023-08-09
>
> We thank Reviewer G2AN for the helpful comments and for providing thoughtful feedback on our work.
>
> ***
>
> **1. Limited novelty**
>
> > The proposed method has limited novelty, and it is a combination of different existing modules. For example, emphasizing the hand-region features and contrastive learning with inverse temporal orders are widely used in many existing works.
>
> Unfortunately the reviewer does not indicate which specific existing work he/she is concerned about.  We would be glad to elaborate with those reference(s) in mind if the reviewer can provide references.
>
> In our related works section, we thoroughly discuss existing techniques and indicate their differences with AE2 (video contrastive learning approaches with inverse temporal orders in Ln 76-85 and works that emphasize the hand-region features in Ln 101-108).
>
> We respectfully disagree with the comment of limited novelty and provide clarifications below. As recognized by other reviewers, in this work, we address a less studied research problem (Reviewer YpGy), and propose a novel self-supervised embedding approach to learn fine-grained view-invariant features (Reviewer CYjc and ghSs).
>
> Our innovation arises from two main aspects:
>
> **Problem Space.** We focus on learning ego-exo view-invariant features from in-the-wild videos that are not captured simultaneously, or in the same environment — a problem that has been unexplored thus far.
>
> **Technical Contributions.** In response to the unique challenges presented by the ego-exo domain, we have developed two techniques: (1) Recognizing the substantial differences between ego and exo views, we devise an object-centric encoder design that emphasizes hand-object interaction regions, effectively bridging the ego-exo domain gap. (2) To address the complexities in aligning different-viewpoint video pairs and the potential temporal variations within a video, we introduce a global perspective for creating negative samples. This contrasts with the frame-by-frame sampling methods used in previous approaches. Both techniques are specifically tailored to our new problem space and have not been previously discussed in the context of fine-grained action feature learning [15, 22, 23, 41, 53, 61, 64].
>
> If Reviewer G2AN knows of specific references that we might have overlooked in our related work, we would gladly examine them and discuss how they differentiate from our research.
>
> ***
>
> **2. Issues of the proposed benchmark**
>
> > (a) Limited samples…CMU-MMAC…This makes the model training easy to overfit. (b) Limited diversity... which casts doubt on the generalizability of the proposed method.
>
> > please elaborate more on the details, such as the number of action classes in each dataset.
>
> We recognize the concerns regarding the dataset scale and diversity in our proposed benchmark. Since the field of ego-exo fine-grained feature learning is new, the development of our benchmark was both an essential and challenging step. Here's how we've approached these issues:
>
> **Comparative Scale and Avoiding Overfitting.** We carefully designed our benchmark to align with the scale of datasets widely adopted by the fine-grained action feature learning community [15, 22, 23, 33, 41, 53], such as:
> + Pouring [63] (70 for training, 14 for validation), 1 action class
> + Penn Action [82] (40-134 for training, 42-116 for validation), 13 action classes
> + IKEA ASM [3] (61 for training, 29 for validation), 1 action class
> + H2O ego-only set [32] (27 for training, 11 for validation), 1 action class
>
> Moreover, unlike these examples that split data only into training and validation, we included a separate test set (Ln 684-685 in the Appendix) to provide a more robust evaluation and mitigate the risk of overfitting. We believe this approach contributes to a more fair and comprehensive assessment within the field.
>
> **Dataset Clarifications and Method Generalizability.** As described in Ln 243-259 in the main paper and Section A.2.1 in Appendix, our benchmark includes four action-specific datasets (not only CMU-MMAC), each linked to one action category (Ln 247 and Ln 608-637), encompassing videos from both controlled and in-the-wild environments. Importantly, the action-specific focus of our benchmark aligns with existing fine-grained action feature learning literature [15, 22, 23, 33, 41, 53]. The emphasis here is on understanding and reasoning about the temporal evolution of one specific action rather than classifying the action category (Ln 18-24). In addition, we would like to emphasize the diversity of these four datasets in our benchmark, and AE2’s superior performance across them. As shown in Table 1, AE2 consistently outperforms all baseline approaches on all datasets; these results offer strong support for AE2’s generalizability.
>
> **Cost constraints.** While our approach is entirely self-supervised and does not require frame-wise labels during training, *evaluating* the learned representations on downstream tasks necessitates per-frame labels for every video in the dataset. We incurred significant expense to create the new frame-labeled ground truth sets for evaluation by us and others down the road.
>
> In conclusion, we believe our benchmark provides a valuable starting point for the emerging field of ego-exo fine-grained feature learning. We appreciate Reviewer G2AN’s suggestions, and will strive to enhance its scale and diversity in future iterations.

---

> > ### Comment · Reviewer_G2AN · 2023-08-20
> >
> > Thank you for the reply!
> >
> > The paper is well-motivated, and this study is essential in video understanding. I also appreciate the authors' efforts in developing the benchmarking datasets, but my concerns about the size of the datasets remain there. Therefore, I will keep my original rating.

---

> > > ### Author Response · Authors · 2023-08-20
> > > **Response to Reviewer G2AN's further comment**
> > >
> > > Thank you for taking the time to read our rebuttal! We are delighted that you recognize our work as well-motivated and essential in the field of video understanding, and we appreciate your acknowledgement of our efforts in developing the ego-exo benchmark. Your concerns regarding the dataset scale are noted, and we'd like to provide additional clarity:
> > >
> > > Below, we list datasets widely adopted by the fine-grained action feature learning community, along with representative works for comparison:
> > >
> > > **Datasets**:
> > > + (a) Pouring: 70 videos for training, 14 videos for validation
> > > + (b) Penn Action: 40-134 videos for training, 42-116 videos for validation
> > > + (c) IKEA ASM: 61 videos for training, 29 videos for validation
> > > + (d) H2O ego-only set: 27 videos for training, 11 videos for validation
> > >
> > > **Representative works**:
> > > + TCC [15], published on CVPR’19, uses 2 datasets: (a) & (b)
> > > + LAV [23], published on CVPR’21, uses 3 datasets: (a), (b) & (c)
> > > + VAVA [41], published on CVPR’22, uses 4 datasets: (a), (b), (c) & COIN (specific subset scale not found in the paper)
> > > + CASA [33], published on CVPR’22, uses 3 datasets: (a), (b) & (d)
> > >
> > > As stated in Ln 243-259 and Table 3 in the Appendix, our ego-exo fine-grained feature learning benchmark aligns with (or exceeds) the scale of existing benchmarks, with each of the four datasets including over 100 videos.
> > >
> > > Though the number of videos might seem small at a glance, it's essential to understand that we are focused on learning **frame-wise** representations. The total number of training samples equals the number of frames, reaching a scale of a few thousands (8k-30k per action class in our case). To offer a robust evaluation and reduce the risk of overfitting, we have included *four* diverse datasets and incorporated *a separate test set*, departing from the common practice of merely splitting videos into train and validation.
> > >
> > > We trust that the information above offers a comprehensive perspective on our dataset scale. Should you have any additional inquiries, please don’t hesitate to share your comments. We will try our best to provide timely responses before the discussion period ends.
> > >
> > > Thank you once again for your thoughtful review!

---

> > > > ### Comment · Reviewer_G2AN · 2023-08-21
> > > >
> > > > Thank you for the reply. It addressed my concerns on the dataset scale.

---

### Official Review · Reviewer_ghSs · 2023-07-05

**Soundness:** 3 good
**Presentation:** 3 good
**Contribution:** 3 good
**Rating:** 6
**Confidence:** 4

**Summary:**

The paper introduces a novel approach for learning view-invariant features from exo and ego views using temporal alignment. The key idea is to utilize unpaired ego-exo data for the pretext task of temporal alignment in a self-supervised setting. Furthermore, an object-centric encoder was used to focus on hand and object regions. The authors also introduce a regularization method to maximize the margin between positive pair and their corresponding negative pairs. The negative pair is constructed by replacing the one video of the pair with its reversed.


**Strengths:**

- The paper is well-written and easy to follow
- The paper introduces a novel framework for view-invariant representation learning
- The proposed approach is capable of utilizing the unpaired data where the videos of a pair do not have to be synchronized or be in the same environment
- The method is compared with a reasonable set of baseline methods



**Weaknesses:**

  - The idea of sequence alignment as self-supervised video representation learning has been studied by e.g. [22, 23]. It is not clear why the methods of [22, 23] cannot be applied to ego-exo data.
- In Line205-211 it is argued that frame-by-frame sampling in contrastive loss can lead to incorrect negative samples due to the periodicity of some videos, however, it is not clear how the proposed approach of using reversed videos for the negative sample does not suffer from periodicity.


**Questions:**

See weakness.

**Limitations:**

Yes, the limitation is addressed.

---

> ### Author Rebuttal · Authors · 2023-08-09
>
> We thank Reviewer ghSs for the helpful comments and for providing thoughtful feedback on our work.
>
> ***
>
> **1. Applicability of existing sequence alignment approaches**
>
> >  The idea of sequence alignment as self-supervised video representation learning has been studied by e.g. [22, 23]. It is not clear why the methods of [22, 23] cannot be applied to ego-exo data.
>
> We would like to clarify that these prior approaches *can* be directly applied to ego-exo data. Specifically, we have provided these comparisons in Table 1 of our manuscript, where we compared a total of 8 baselines (including two sequence alignment approaches: TCC [15] and GTA [22]) with AE2. Note that LAV [23] shares similarities with GTA in terms of the dynamic time warping (DTW) alignment loss, albeit with a different regularization term. Since LAV [23] does not have publicly released code, we implemented GTA [22] using their officially released code in our experiments.
>
> From Table 1, we observe the inferior performance of these baselines compared with AE2, since these sequence alignment approaches are developed for third-person videos only and do not account for the ego-exo viewpoint discrepancy. Contrarily, our proposed AE2 effectively addresses the unique challenges in the ego-exo domain, outperforming all baselines across a wide range of downstream tasks. These results substantiate our claim that AE2 is superior in learning fine-grained ego-exo view-invariant features.
>
> ***
>
> **2. Sequence-level negatives vs. frame-level negatives**
>
> > it is not clear how the proposed approach of using reversed videos for the negative sample does not suffer from periodicity.
>
>
> We thank Reviewer ghSs for this insightful question, and clarify below how our proposed sequence-level negatives (using reversed videos) can better address the periodicity within a video sequence.
>
> As a motivating example, Figure R2 (available in the rebuttal PDF) depicts a video sequence of breaking eggs with a certain amount of periodicity (where the camera wearer hits the egg to the bowl’s edge twice). Conventional frame-by-frame sampling approaches [10, 61, 64] treat temporally close frames as positive pairs and temporally distant ones as negative pairs. As a consequence, the two visually similar frames (marked by a red border) would be incorrectly identified as a negative pair by those existing methods. Contrarily, our method of constructing negative samples adopts a global perspective, addressing temporal fluctuations within a video more effectively. By reversing the entire sequence, we create a challenge in aligning the reversed ego view video with the original exo view video, thus forming a valid negative sample.
>
> This methodology is not only conceptually appealing but also empirically validated. Table 1 in the main paper illustrates the effectiveness of our proposed AE2, demonstrating that it outperforms prior techniques utilizing frame-wise negatives [10, 61, 64]. These results offer compelling empirical evidence that our sequence-level negative sampling strategy successfully mitigates the issues related to periodicity and improves performance, affirming its superiority over conventional frame-by-frame sampling methods.

---

> > ### Comment · Reviewer_ghSs · 2023-08-21
> >
> > My concerns have been resolved and I appreciate the authors for the clarification.

---

### Official Review · Reviewer_CYjc · 2023-07-06

**Soundness:** 3 good
**Presentation:** 3 good
**Contribution:** 3 good
**Rating:** 5
**Confidence:** 5

**Summary:**

The authors proposed a novel self-supervised embedding approach AE2 to learn fine-grained view-invariant frame-wise video features from unpaired egocentric and exocentric videos. AE2 consists of two core designs: (1) an object-centric encoder that explicitly focuses on
objects; (2) a contrastive-based alignment objective that leverages temporally reversed frames as negative samples. Four datasets exhibit improvement in experiments, demonstrating a feasible framework for learning fine-grained view-invariant features.

**Strengths:**

1. This paper presents a well-motivation idea and sound arguments in general. It makes sense to learn view-invariant video features form unpaired data.
2. Ablation studies are conducted to evaluate the effectiveness of each design.
3. The visualization of learned view-invariant video features for test videos is very impressive, which demonstrates effectively capture the progress of an action while remaining view-invariant.

**Weaknesses:**

1. In Lines 225-226 “ In creating negative samples, we opt for reversing the frames in S rather than randomly shuffling them.” Although the authors explain the superiority of the reversing frames, authors need to provide further experiments comparing reversing frames with randomly shuffling them.
2. Baseline's experimental results are so good that the baseline achieves SOTA results on most of the datasets. why is the baseline's experimental performance so good?
3. Lack of analysis of experimental results in the ablation experiment section.
4. In Training and Implementation Details, “the object-centre encoder network is optimized to minimize the loss above for 240 all pairs of video sequences in the training set, including ego-ego, exo-exo, and ego-exo pairs.”  The training process introduces data from the same viewpoint and paired data, but the network structure is designed specifically for unpaired data from different viewpoints. Can separate experiments be performed only on unpaired data from different viewpoints to demonstrate the validity of the proposed method?
5. The language should be further improved by correcting some language errors, such as “ See Supp. for full implementation deatils ...” (Lines 232 in the paper).

**Questions:**

See Weaknesses

---

> ### Author Rebuttal · Authors · 2023-08-09
>
> We thank Reviewer CYjc for the helpful comments and for providing thoughtful feedback on our work.
>
> ***
>
> **1. Comparing temporally reversing frames with randomly shuffling frames**
>
> Please refer to our general response.
>
> ***
>
> **2. Baseline’s experimental results**
>
> > Baseline's experimental results are so good that the baseline achieves SOTA results on most of the datasets.
>
> We appreciate your observation regarding the Base DTW row in Table 2. Base DTW is indeed a strong baseline, as it adopts dynamic time warping (DTW) loss [4] to align video pairs. Specifically, we implemented a smooth differentiable version of the DTW loss, positioning Base DTW as a variant of the SOTA baseline GTA [22]. This strong design foundation and the well-established efficacy of DTW in video self-supervised learning [6, 8, 22, 23, 77] contribute to Base DTW's robust performance.
>
> Furthermore, when considering the performance of Base DTW, it is useful to examine both Table 2 in the main paper and Table 8 in the Appendix — the latter being a complete version of Table 2. While Base DTW ranks strongly on the action phase classification task in Table 2 (i.e., 5th on dataset (A), 2nd on datasets (B) and (C), and 3rd on dataset (D)), its performance is more moderate when observed across a broader range of evaluations in Table 8. For instance, it ranks 4th on (A), 3rd on (B), 7th on (C), and 3rd on (D) in ego2exo frame retrieval.
>
> While acknowledging the strength of Base DTW, it's essential to highlight that our proposed AE2 consistently outperforms it. In both Table 2 & 8, AE2 secures the top position across various evaluations.
>
> We hope the discussion above clarifies why Base DTW exhibits good performance in specific instances and provides a balanced view of its overall capabilities.
>
> ***
>
> **3. Lack of analysis of experimental results in ablation**
>
> Due to space constraints, we offer a brief discussion of our ablation study in the main paper (Ln 301-311) and a more comprehensive analysis in the Appendix (Ln 727-733). Our findings highlight that the object-centric encoder greatly boosts downstream task performance. This boost can be attributed to the encoder's emphasis on shared information between ego and exo views (i.e., hand-object interaction regions), which effectively bridges the ego-exo domain gap. Furthermore, contrastive regularization amplifies the quality of the learned representations by creating valid negative sample pairs, leading to additional improvements.
>
> If there are particular aspects or questions that Reviewer CYjc would like us to elaborate on, we would be more than happy to address them.
>
> ***
>
> **4. Experiments on unpaired data from different viewpoints**
>
> > The training process introduces data from the same viewpoint and paired data, but the network structure is designed specifically for unpaired data from different viewpoints.
>
> In general, unpaired and different-viewpoint video pairs pose more complex constraints than their counterparts (i.e., paired, same-view) data. With this understanding, we designed AE2 to be versatile enough to accommodate all scenarios (i.e., paired same-view, unpaired same-view, paired different-view, and unpaired different-view). As we detail below, AE2 demonstrates consistently superior performance across these cases, outperforming baseline approaches.
>
> **Paired & Unpaired Data.** AE2 is designed to encompass a diverse set of training scenarios. While the first two datasets (CMU-MMAC and H2O) include a proportion of paired (ego, exo) videos, the latter two (Pour Liquid and Tennis Forehand) contain naturally unpaired videos. Note that even for the first two datasets, we never utilize the ego-exo synchronization as an explicit supervision signal in the training phase (unlike baselines [61, 64], which *do* directly exploit the pairing information). Rather, we treat all pairs as if they are unpaired (Ln 256). As shown in Table 1 of our manuscript, we observe superior performance of AE2 across all four datasets, indicating its generalizability on both scenarios.
>
> **Same-Viewpoint & Different-Viewpoint.** Including both same-viewpoint and different-viewpoint video pairs in our training data is a deliberate choice to foster generalizability. Considering $n_1$ ego videos and $n_2$ exo videos, different-viewpoint pairs give $n_1 \times n_2$ data, while incorporating all pairs grows this to $(n_1 + n_2) \times (n_1 + n_2)$ training data. This expansion ensures the model learns meaningful representations that can temporally align both types of videos, enhancing its flexibility. As shown in our supplementary video (Ln 320-322), AE2 succeeds in aligning not only ego-exo (different-viewpoint) videos but also unpaired ego or exo (same-viewpoint) videos; this lends support for the effectiveness of including same-viewpoints training pairs in AE2.
>
> **Separate Experiment on Unpaired Data from Different Viewpoints.** In line with Reviewer CYjc’s suggestion, we conducted focused experiments on the (unpaired, different-viewpoint) setting — using only different-viewpoint data pairs for training on the two naturally unpaired datasets, Pour Liquid and Tennis Forehand. The results reveal that when applied to cross-viewpoint pairs, AE2 consistently outpaces the strongest baseline GTA [22]. Specifically, for Pour Liquid, the comparison between GTA (cross-viewpoint pairs) and AE2 (cross-viewpoint pairs) yielded F1 scores of 63.24 vs 63.80 and mAP@10 scores of 59.17 vs 62.73. Similarly, for Tennis Forehand, the F1 scores were 84.23 vs 84.48 and mAP@10 scores were 86.38 vs 87.17. These results further validate AE2’s ability to adeptly handle the challenges posed by unpaired data from different viewpoints.
>
> We thank Reviewer CYjc for this insightful question, and will include the discussion and additional experiments in the manuscript.
>
> ***
>
> **5. Language error “deatils’’**
>
> We will thoroughly review and correct this typo and any other language errors in our manuscript.

---

### Official Review · Reviewer_YpGy · 2023-07-06

**Soundness:** 3 good
**Presentation:** 3 good
**Contribution:** 3 good
**Rating:** 6
**Confidence:** 4

**Summary:**

The manuscript proposes a self-supervised learning approach for view-invariant representation learning from egocentric and exocentric videos. Unlike prior approaches, the proposed method is applicable to videos that are not aligned and captured in different environments. The key idea is to learn the alignment between the frames from two views using contrastive learning by using the temporally reversed version of the video as the negative sample. The manuscript also develops an object-centric encoder that focuses on the hands and active object regions in the frames. This allows the model to ignore the less informative regions such as background. The manuscript also develops a benchmark for evaluating the effectiveness of view-invariant representation approaches. Comparison of the proposed approach on the new benchmark with prior approaches clearly reveal its effectiveness.

**Strengths:**

The manuscript addresses a problem that is not widely studied. Unlike prior view-invariant learning approaches that sets hard constraints such as availability of time-synchronized videos or videos captured in the same environment, the proposed approach can be successfully applied to unpaired videos captured at different environments.

The benchmark developed could also be useful to the research community.

The manuscript is written in a clear way explaining the various motivations behind the key ideas and the methodology used.

The manuscript also presents an extensive set of experiments to validate its effectiveness by comparing against existing approaches. Moreover, most of the design choices adopted in the proposed approach are also validated with empirical analysis.


**Weaknesses:**

One of the contributions of the manuscript is the use of time reversed frames as the negative sample for contrastive learning instead of random shuffling. However, there is no experimental validation of the proposed negative sampling strategy against random shuffling.

Continuing from the above, the proposed negative sampling strategy could affect actions that are constituted of repetitive subactions such as cutting something, playing drums, etc. where the time reversed frames could look more or less similar to its original version. The authors mention in the supplementary that for videos containing non monotonic actions (pouring water) only the first/last half of the frames are reversed. However, it is not clear even such an approach would work on repetitive actions. It seems the proposed negative strategy will work only on a selected set of actions.


**Questions:**

Please see the weaknesses section for the main queries.

The learnable spatial embedding $\textbf{e}_{spatial}$ is mentioned as the embedding to encode the bounding box coordinates and the confidence score obtained from the hand-object detector. It is not clear how this is obtained for the global image feature $\phi _{project_g}(g_i)$.

Some implementation details such as how the base encoder is initialized and whether it is frozen/finetuned during training are missing.


**Limitations:**

The manuscript has adequately addressed the limitations.

---

> ### Author Rebuttal · Authors · 2023-08-09
>
> We thank Reviewer YpGy for the helpful comments and for providing thoughtful feedback on our work.
>
> ***
>
> **1. Comparing temporally reversing frames with randomly shuffling frames as the negative sample**
>
> > no experimental validation of the proposed negative sampling strategy against random shuffling
>
> Please refer to our general response.
>
> ***
>
> **2. Negative sampling strategy**
>
> > the proposed negative sampling strategy could affect actions that are constituted of repetitive subactions such as cutting something, playing drums, etc. where the time reversed frames could look more or less similar to its original version
>
> We agree that periodic actions present specific challenges, as we discussed in Ln 208-211.  However, we would like to emphasize that our negative sampling strategy offers a more robust solution to tackle the intricacies of repetitiveness within an action, when compared with frame-by-frame sampling approaches adopted in prior works [10, 61, 64].  As noted in Ln 223-228, instead of creating negatives at the frame level, we opt for a more holistic, global perspective to construct negative samples, yielding more robust negatives and stronger representations.
>
> **Qualitative Insights.** Figure R2 (available in the rebuttal PDF) depicts a video sequence of breaking eggs with a certain amount of periodicity (where the camera wearer hits the egg to the bowl’s edge twice). Frame-by-frame sampling approaches [10, 61, 64] treat temporally close frames as positive pairs and temporally distant ones as negative pairs. As a consequence, the two visually similar frames (marked by a red border) would be incorrectly identified as a negative pair by those existing methods. Contrarily, our method of constructing negative samples adopts a global perspective, addressing temporal fluctuations within a video more effectively. By reversing the entire sequence, we create a challenge in aligning the reversed ego view video with the original exo view video, thus forming a valid negative sample.
>
> **Quantitative Evidence.** Beyond this illustrative example, Table 1 in the main paper demonstrates the efficacy of our proposed AE2. It showcases the leading performance of AE2 compared to prior techniques that utilize frame-wise negatives [10, 61, 64]. These results provide strong empirical evidence for the superiority of our proposed negative sampling strategy.
>
> **Clarification on Training Setting.** It’s pivotal to understand the context in which AE2 operates. Consistent with current fine-grained action feature learning approaches [15, 22, 23, 41, 53],  AE2 focuses on videos portraying a single action instance (e.g. one instance of breaking an egg) rather than videos involving multiple takes of the same action (e.g., breaking multiple eggs), as highlighted in Ln 57 and Ln 136-138. Within this context, our negative sampling technique excels in generating valid negative samples and proves its mettle against the challenge of repetitive and background frames. Finally, it remains an open question in the community to generalize existing techniques to longer videos featuring multiple occurrences of the same action. Addressing this challenge remains on our horizon for future work, as indicated in Ln 336-338.
>
> ***
>
> **3. Implementation details: spatial embedding and base encoder initialization**
>
> We appreciate your attention to the details. Concerning the learnable spatial embedding $e_{spatial}$, it is designed for local image features and not incorporated in global feature computation. We will rectify the equation in Ln 187 of our manuscript to exclude $e_{spatial}$.
>
> Regarding the base encoder, it is initialized with a ResNet-50 pretrained on ImageNet, and is jointly optimized along with the transformer encoder throughout the training process. In addition, the architecture of our proposed encoder is detailed in Ln 240-328 within models/embedder.py in the submitted Appendix zip.
>
> We will include these details in our updated manuscript, and pledge to open-source our code for complete reproducibility.

---

> > ### Comment · Reviewer_YpGy · 2023-08-19
> > **Response to authors**
> >
> > I appreciate the authors' efforts in providing a detailed response. All the concerns raised in the initial review stage are successfully addressed in the rebuttal. So I am keeping my initial rating.

---

> > > ### Author Response · Authors · 2023-08-20
> > > **Response to Reviewer YpGy**
> > >
> > > Thank you for your insightful feedback and valuable support. We are pleased to confirm that we have fully addressed your concerns. In light of your thoughtful suggestions, we will make the necessary updates to the manuscript.

---

### Author Rebuttal · Authors · 2023-08-09

We thank all reviewers for their thoughtful and constructive review of our manuscript.

Three of the four reviewers recommend accepting.  We were encouraged to hear that the reviewers found our work novel (CYjc, ghSs), sound (Cyjc), well-motivated (CYjc, G2AN), backed up with extensive experiments (YpGy, G2AN), detailed ablation studies (YpGy, CYjc), and compelling visualizations (CYjc); and that they view our proposed ego-exo benchmark to be new (G2AN) and useful to the community (YpGy).

In response to the feedback received, we provide a general response here to address a query shared by two reviewers, and individual responses below to specific points from each reviewer.  Please refer to the attached rebuttal PDF, where supplementary figures and tables have been provided to substantiate our responses.
***
**1. Comparing temporally reversing frames with randomly shuffling frames as the negative sample**

We thank Reviewer YpGy and CYjc for raising this point. In compliance with their suggestions, we conducted additional experiments to validate the efficacy of creating negative samples by temporally reversing frames, as opposed to random shuffling. Table R1 (available in the rebuttal PDF file) compares these two frame sampling approaches across all four datasets.

The results in Table R1 reveal that, in general, temporally reversing frames yields superior and more consistent performance than randomly shuffling. For example, on (A) CMU-MMAC data, random shuffling results in a decrease of -5.62% in F1 score and -4.19% in mAP@10 compared with regular AE2.


As explicated in Ln 223-228 of our manuscript, the inferior performance of random shuffling can be related to the abundance of similar frames within the video sequence. Even after shuffling, the frame sequence may still emulate the natural progression of an action, thereby appearing more akin to a positive sample. Conversely, unless the frame sequence is strictly symmetric (a scenario that is unlikely to occur in real videos), temporally reversing frames is apt to create a negative sample that deviates from the correct action progression order.

To further illustrate this point, we visualize two video sequences in Figure R1 (available in the rebuttal PDF file). From this figure, it is evident that the randomly shuffled sequence seems to preserve the sequence of actions like breaking eggs or pouring liquid, thereby resembling a “positive” example. Consequently, incorporating such negative samples into training may confuse the model and lead to the observed diminished performance.

Going back to our results in Table R1, we can dissect the performance gains further. The large improvement observed on dataset (A) when using temporally reversed frames over randomly shuffled frames stems from its inherent ability to be more robust to the repetitive nature of the breaking egg action (e.g., the camera wearer may hit the egg twice before breaking it, resulting in many similar frames in the video sequence). On the other hand, the subtler improvement on dataset (D), dealing with the tennis forehand action, is likely due to the strictly monotonic nature of this action, leaving less room for our method to outperform. However, it is essential to emphasize that this does not diminish the value of our approach, our negative sampling strategy still proves preferable consistently across all four datasets.

We will include the new baseline, the corresponding figure and a detailed discussion regarding these two frame sampling strategies in the manuscript.
***
We would again like to thank all reviewers for their time and feedback, and we hope that our responses adequately address all concerns. Any further questions are highly welcomed.

---

### Decision · Program_Chairs · 2023-09-21

**Decision:**

Accept (poster)

**Comment:**

All reviewers were in agreement, recommending acceptance. The AC reviewed the paper, reviews, and rebuttals, and agrees with the consensus. The paper presents a novel approach to view-invariant representation learning, addressing a unique problem not widely studied before - unlike previous methods that rely on stringent data constraints, this approach is adaptable to unpaired videos from diverse environments. The experimental analysis provides robust validation of the method's efficacy. The introduced benchmark can also be a significant contribution to the community. Important concerns from reviewers were resolved during the rebuttal period. Please, ensure that the new figures, baselines, and other clarifications provided in the rebuttal are added to the main manuscript.